# *Plantae tinctoriae*: The 1759 Dissertation on Dye Plants by Engelbert Jörlin

**Regina Hofmann-de Keijzer [1,*,†] and Matthijs de Keijzer [2,†]**

[1]   Institute of Art and Technology, Conservation Science, University of Applied Arts Vienna,
      1010 Vienna, Austria
[2]   Cultural Heritage Laboratory, Cultural Heritage Agency of the Netherlands, Hobbemastraat 22,
      1071 ZC Amsterdam, The Netherlands
[*]   Correspondence: regina.hofmann@uni-ak.ac.at
[†]   The authors are retired now.

**Abstract:** In the late 1750s, the Swedish botanist Engelbert Jörlin (1733–1810), one of Carl Linnaeus' students wrote his dissertation *Plantae tinctoriae* on more than one hundred dye plants. The article presents a systematic study on these dyeing materials and reflects the knowledge in the mid-18th century. His dissertation focused on domestic plants that could be suitable instead of expensive imported trade goods and was published during the Age of Utility (1719–1771). The Latin text of Jörlin's dissertation was first converted into a digital version by the 'Noscemus General Model' from Transkribus and then translated into English. The current scientific names were obtained from various biological websites. The dyestuffs were assigned to four groups: native and applied in Sweden (A); imported trade products (B); native to Sweden with potential use for dyeing (C); non-native and used abroad (D). They were mainly applied for dyeing textiles, less frequently for pharmaceuticals, cosmetics (make-up), inks and artists' pigments. In his dissertation, Jörlin refers to scriptures from antiquity, Latin botanical literature from the 16th and 17th centuries but especially to the publications of Carl Linnaeus.

**Keywords:** Engelbert Jörlin; Swedish dye plants; traded dyeing materials; Carl Linnaeus; Age of Utility

## 1. Introduction

The famous Swedish botanist, zoologist and physician Carl Linnaeus (1707–1778) published 185 dissertations by his students in his 10-volume work of *Amoenitates Academicae* (1749–1790) [1,2], including Engelbert Jörlin's thesis in the fifth volume from 1760 [3]. At the beginning of the 19th century, in 1813, the Austrian naturalist Johann Georg Megerle von Mühlfeld (1780–1831) wrote his book on Austrian dye plants and cited Jörlin's dissertation as "*Linné Dissertatio de plantis tinctoriis in den Amoen. Academ. V. Theil. S. 314*" [4].

In 1759, Jörlin's thesis was published in Sweden during the Age of Utility (Age of Freedom, 1719–1771) [5]. At that time, domestic products were encouraged and sciences, which supported the Swedish economy received subsidies. The research focused on finding indigenous plants and animals that could be substitutes for expensive imported dyeing goods. Pioneers in the field of Swedish dye plants are the botanist Johan Linder (1676–1724) with his 1720 publication *Swenska Färge-Konst* (*Swedish Dye Craft*) and Linnaeus, who describes many dye plants in his scientific writings.

The article presents 130 materials from dye plants and dye insects, reflecting the state of knowledge about domestic and traded dyeing products in the mid-18th century. Furthermore, the publication gives an overview of Engelbert Jörlin's life, followed by a general description of his dissertation. The dyeing materials are discussed according to the

colors given by Jörlin and the last chapter is dedicated to lichens, which were of particular interest during the Age of Utility in Sweden. To explain the dyeing properties, current scientific literature served to classify the dyeing materials into chemical dye classes such as flavonoids, anthraquinones, indigoids and tannins [6–8]. The information of Jörlin's dissertation is presented in the present tense without quoting Jörlin's name each time, and the remaining information is written in the past tense. At the end of each chapter, a table provides information on the dyeing materials, the numbers used by Jörlin, the names by Jörlin and Linnaeus and their current scientific names.

## 2. Engelbert Jörlin's Life

Engelbert Jörlin was born in Jörlanda, Västra Götaland County, Sweden on the 8th of June 1733 [9]. He was the second eldest of ten children of the farmer Sven Börjesson (1703–1757) and his wife Börta Engelbrektsdotter (1709–1797). During a parish visit, young Engelbert was noticed by Bishop Georg Wallin, who took him as his foster son and let him study. From the 20th of September 1757, he studied at the Uppsala University, where he became one of Linnaeus' most talented students. In 1759, Jörlin defended his dissertation *Plantae tinctoriae* in front of his supervisor Linnaeus and became Magister of Philosophy in 1761 (Figure 1).

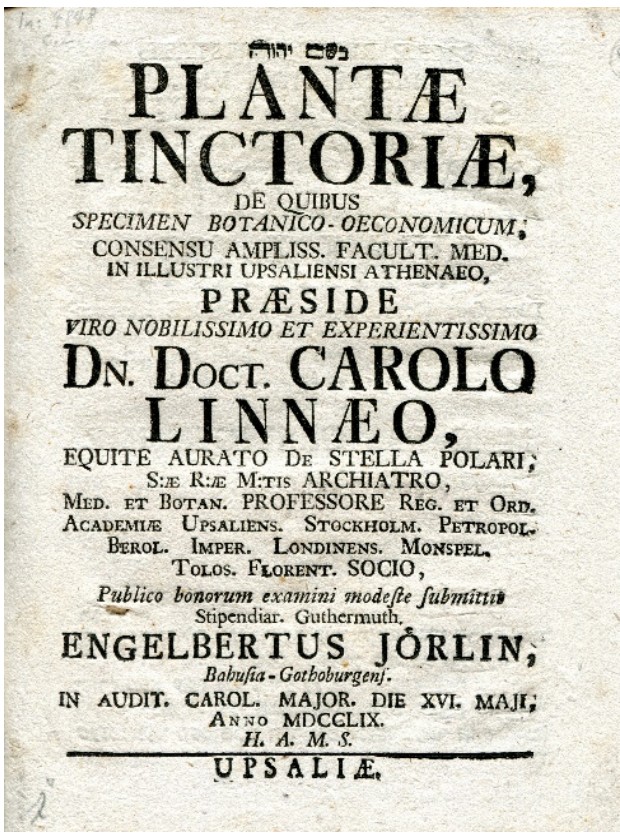

**Figure 1.** Engelbert Jörlin's dissertation. Image: Courtesy of Hunt Institute for Botanical Documentation, Carnegie Mellon University, Pittsburgh, PA, USA.

A group of students, named the 'Apostles' of Linnaeus, undertook botanical and zoological expeditions to various parts of the world. Unfortunately, nearly half of them died during their journey [10]. After graduating, Jörlin also wanted to become an Apostle intending to study the flora and fauna of South Africa. However, the Dutch did not give him permission to travel to the Cape. He then wanted an employment at the Dutch East India Company, warmly recommended by Linnaeus. Again, he was unsuccessful because

he did not have a full medical education. Despite all efforts, Jörlin did not become one of Linnaeus' Apostles. Both rejections may have had a positive influence on his lifespan.

After completing several scientific journeys through Sweden, Denmark and northern Germany, he became docent in natural history at the Lund University in 1769 and associate professor (Swedish: extra ordinarie adjunct) in 1781. He also lectured at the gymnasium in Gothenburg; in 1784, he was appointed rector of the Trivial School in Gothenburg, a post he held for several years. In addition to his teaching activities, Jörlin devoted himself to the cultivation of Swedish plants, which is reflected in his publications *Flora macelli hortensis—Svenska Koks-och Kryddgården* (Flora macelli hortensis—Swedish Kitchen and Herb Garden, 1784) and *Svenska vilda träds och buskars plantering* (Planting of Swedish Wild Trees and Shrubs, 1801) (Figure 2). The 2nd and 3rd volume of the *Flora macelli hortensis* were published in 1786 and 1794, and a joint edition was issued in 1796 [11] (p. 288). On the 27th of May 1787, he married the painter Christina Elisabeth Carowsky (1745–1797) and they had one daughter, Britta Maria Jörlin (1789–1848). Engelbert Jörlin passed away on the 20th of June 1810 in Gothenburg.

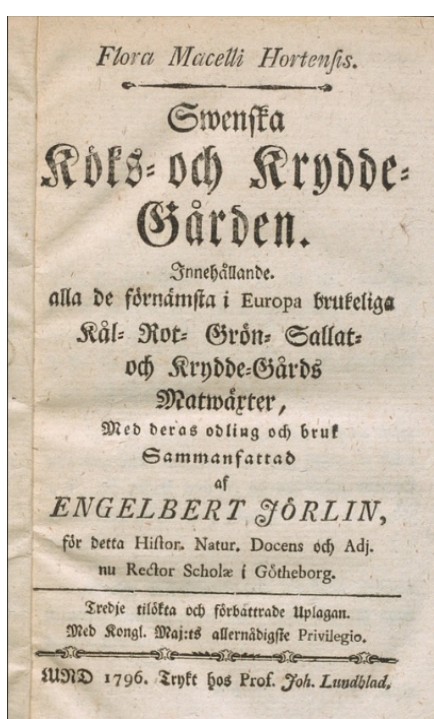

**Figure 2.** *Flora macelli hortensis*—Swedish Kitchen and Herb Garden (1796) [12]. Swedish Literature Bank https://litteraturbanken.se/om/english.html (accessed on 23 November 2022).

### 3. Research Aim and Investigation Methods

The aim of the research was to study Jörlin's dissertation from various points of view and to make this historical document accessible to various disciplines, such as biology, art history, dye history and heritage science. The following investigation methods were performed: first, a digital version of the Latin text was obtained by the 'Noscemus General Model' from Transkribus [13]. The Latin text was then translated into English. The next step was to create a Microsoft Access database with fields for the plant names, Linnaeus' plant systematic and the distribution and application of the dyestuffs. Jörlin took over the scientific names and the classification of the plants from Linnaeus. Linnaeus' taxonomy and classification has evolved, leading to changes in scientific names of plants and animals and their categorisation into specific families. Several websites served to find out the current scientific names, family names and information about the species' distribution: the International Plant Name Index [14], the Plants of the World Online [15], the Linnaean Plant Name Typification Project [16] and the Global Biodiversity Information Facility

websites [17]. Finally, queries were made concerning color and use. One topic of this study was the classification of the dyeing materials into groups according to their occurrence, trade and use (Figure 3).

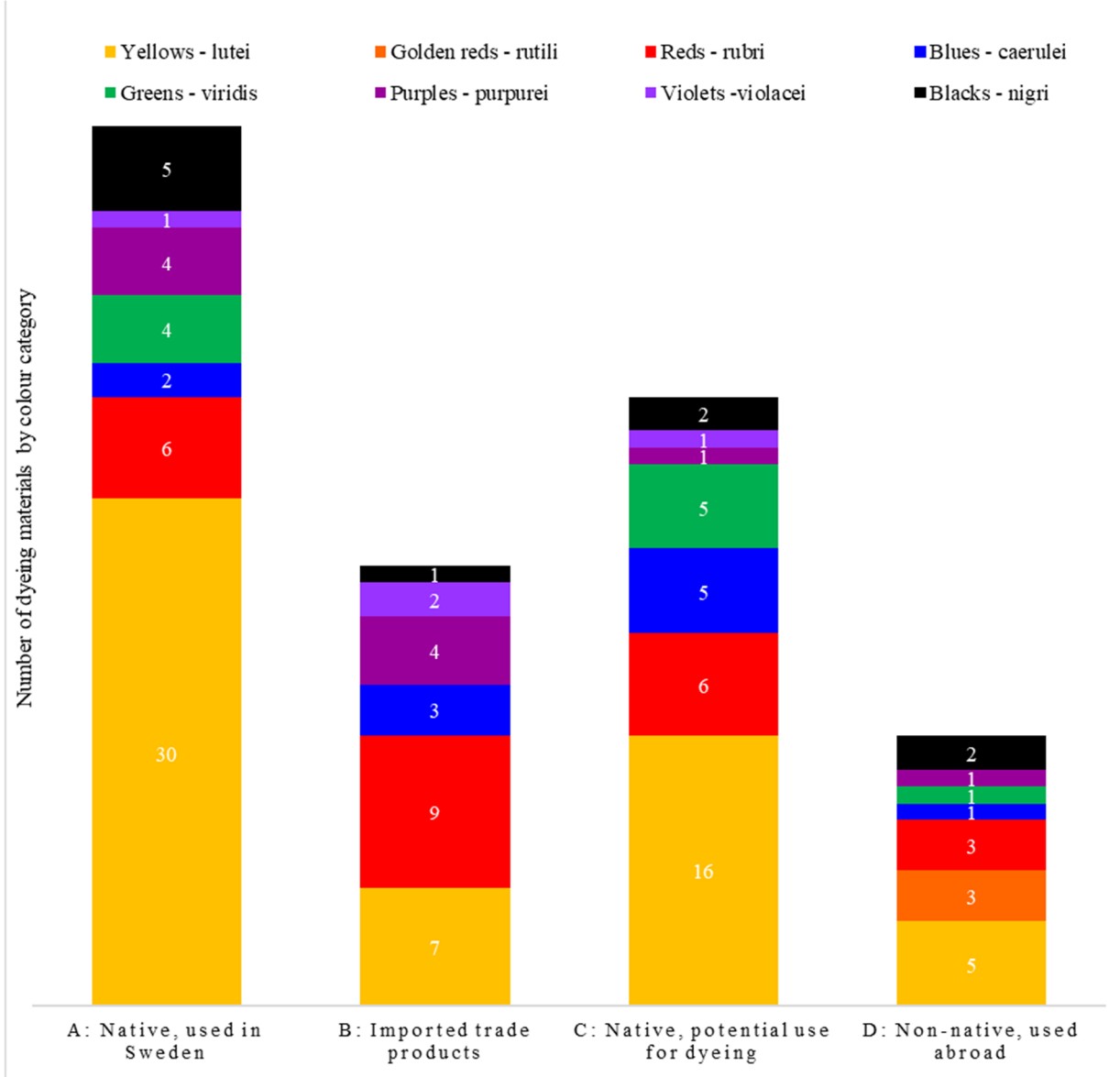

**Figure 3.** The 130 dyeing materials, sorted into four groups according to their occurrence, trade and use.

### 4. Jörlin's Dissertation

Jörlin's thesis gives information on 130 dyeing materials, originating from 117 different plant and animal species assigned to 107 numbers. In the 5th volume of Linnaeus' *Amoenitates Academicae,* the dissertation includes an index of only 102 dyeing materials (Figure 4). Some numbers differ, as can be seen from the asterisked numbers in the Tables 3–14.

The majority of the plant species belongs to seed plants (Spermatophyta). Further, Jörlin mentions one clubmoss, seven lichens, molluscan purple, six gall insects and five dye insects: kermes, Polish cochineal, American cochineal, Indian lac and an indigenous insect found on elm trees. Some species provide more than one dye source, so he lists a total of 130 different dyeing materials: 52 from native species used in Sweden, 26 imported products, 36 materials of native origin having the potential to be used in dyeing and 16

materials of non-native species applied abroad (Figure 3). The bar chart shows that the color yellow dominates among the Swedish dye sources, while the majority of traded products are for dyeing red and purple.

342    PLANTÆ TINCTORIÆ.

| LUTEI. | | RUBRI. | | CÆRULEI. | |
|---|---|---|---|---|---|
| Lign. Santalum | 31. | R. Rubia | 12. | Isatis | 58. |
| Alcanna | 32. | Galium | 10,11. | Indigo | 59. |
| Morus | 79. | Asperula | 9. | Galega | 60. |
| Cort. Rhamnus. | 19,20,21. | Lithosp. | 15. | Croton | 85. |
| Rhus | 24. | Rhus | 25. | Fraxin. | 90. |
| Berberis | 27. | Acetosa | 29. | Delphin. | 51. |
| Prunus | 42. | Torment. | 46. | Campan. | 17. |
| Malus | 43. | Comarī | 47. | | |
| Carpinus | 84. | Lig. Brasilia | 35. | | |
| Rad. Curcuma | 1. | | | | |
| Rumex | 30. | | | VIOLACEI. | |
| Thalict. | 53. | Succ. Sang. Drac. | 28. | Campech. | 34. |
| Urtica | 80. | Cathecu | 98. | Viola | 74. |
| Herb. Luteola | 41. | Bixa | 50. | Satyrium | 75. |
| Serratula | 66. | Cuscuta | 13. | Empetrū | 87. |
| Hieracium | 65. | Phytol. | 42. | | |
| Acanthus | 57. | Basella | 26. | | |
| Bidens | 38. | Inf. Lacca | 102. | | |
| Xanthium | 81. | Carmin. | 103. | | |
| Datisca | 89. | Coccin. | 104. | | |
| Lycopod. | 91. | Polon. | 105. | | |
| Myrica | 88. | Ulmi | 106. | | |
| Salix | 86. | | | | |
| Betula | 76,77. | PURPUREI. | | VIRIDES. | |
| Stachys | 56. | Ligu. Brasiletta | 36. | Senecio | 39. |
| Jacea | 71. | Sappan | 37. | Chæroph. | 22. |
| Polygonū | 33. | Ferneboc. | 38. | Bromus | 6. |
| Lysimach. | 16. | Origani | 55. | Arundo | 7. |
| Succisa | 8. | Roccellæ | 92. | Trifoliū | 63. |
| Anthyllis | 61. | | 93. | Iris | 4. |
| Musci Lichenes | 94-97. | | 94. | Anchusa | 14. |
| Fl. Buphthal. | 70. | | | Pulsatil. | 52. |
| Chæroph. | 22. | RUTILI. | | | |
| Thapsia | 23. | Lawson. | 32. | NIGRI. | |
| Genista | 62. | Fl. Cartham. | 67. | Qverc. | 82, 83, |
| Hyperic. | 64. | Crocus | 5. | | 100. |
| Calendul. | 73. | Gal. Pistac. | 108. | Lycopus | 3. |
| Caltha | 54. | | | Actæa | 48. |
| Succus Gummutta | 49. | | | Melof. | 40. |
| | | | | Genipa | 17. |
| | | | | Uva ursi | 39. |

❋    ❋    ❋

XCIV.

**Figure 4.** Index of dyeing materials published in Linnaeus' *Amoenitates Academicae,* 1760 vol. 5, p. 342 [3]. Digitalisierungsplattform der Zentralbibliothek Zürich e-rara.ch.

The thirty pages of his dissertation begin with religious words of gratitude in Hebrew, which is not found in the 1760 publication. It should be noted that Hebrew, Greek and Latin were taught in Swedish gymnasia. The Old Testament had to be read in Hebrew, the New Testament in Greek [18]. The following text is philosophical and addresses the wonders in nature we perceive with our sensory organs, like the tongue, nose, ears and eyes. The peacock (*Pavo cristatus* Linnaeus, 1758) and the amaryllis (*Amaryllis* L.) are his examples of color variation, and biblical quotations underline his treatise on the beauty and divinity of colors. The next pages are dealing with the aim of the thesis, explanations of terms, physics of colors, the creation of colors by mixing and a general part on the studied plants and animals. Jörlin obtained his information on dye plants from dyers, from his own experiments and from literature. He distinguishes between painting (*pictoriam*) and dyeing (*tinctoriam*). In painting, the pigments are applied on the surface and in dyeing, the dyes are soaked up by the fibres. The dyeing (*tingere*) of wool and silk is performed in the dye works (*infectorium, infectorii*).

He does not discuss the preparation of dyes, as there are already many library books published on this topic. He gives an overview of the common dye sources, most of them originate from the flora, some from the animal world but none from the kingdom of stones. He consulted dyers, lexica and many books of famous authors faced with countless names and general descriptions in many different languages. Jörlin obtained the

information from ancient scriptures, Latin botanical literature from the 16th and 17th centuries but especially from Linnaeus' works in Swedish and Latin, including Linnaeus' 1758 publication on the journey to the Spanish countries in Europe and America by his Apostle Pehr Löfling between 1751 to 1756, who died in Venezuela. The literature is quoted in the form of abbreviations (Table 1).

**Table 1.** Literature cited by Engelbert Jörlin.

| Author | Publication | Year | Language | Abbreviation |
|---|---|---|---|---|
| Publius Vergilius Maro (Virgil) | *Aeneid* | 29–19 BCE | Latin | Virg.aen. |
| Pedanius Dioscorides | *De materia medica* | 50–70 | Greek | Dioscor. |
| Pliny the Elder | *Naturalis Historia* | 77 | Latin | Plin. |
| Johan Bauhin and Johann Heinrich Cherler | *Historia Plantarum Universalis* | 1650–1651 | Latin | Bauh. hist., Bauh. pin. |
| John Ray | *Historia Plantarum* | 1686, 1688 | Latin | (J.) Raj. hist. |
| Johan Linder (Lindestolpe) | *Swenska Färge-Konst* | 1720 | Swedish | Lind. tinct. |
| Georg Eberhard Rumpf (Rumphius) | *Herbarium amboinense* | 1741–1750 | Latin | Rumph. Amb. |
| Pehr Kalm | *Förtekning på någre Inhemska Färge-Gräs* | 1745 | Swedish | Act. Stockh. 1745 |
| Pehr Adrian Gadd | *Försök til en oeconomisk beskrifning öfwer satacunda häraders norra del* | 1751 | Swedish | Gadd. Satag. |
| Carl Linnaeus | *Förtekning, Af de Fargegras, som brukas på Gotland ock Oeland* | 1742 | Swedish | Act. Stockh. 1742 |
| Carl Linnaeus | *Systema Naturae* | 1735 | Latin | Syst. Nat. |
| Carl Linnaeus | *Flora Lapponica* | 1737 | Latin | Fl. Lapp. |
| Carl Linnaeus | *Öländska och Gothländska Resa* | 1745 | Swedish | It. Oel., It. Gothl. |
| Carl Linnaeus | *Flora Suecica* | 1745 | Latin | Fl. svec, Fl. sv. |
| Carl Linnaeus | *Fauna Suecica* | 1746 | Latin | Fn. svec. |
| Carl Linnaeus | *Wästgöta Resa* | 1747 | Swedish | It. W. goth., Itin. W. Gothico |
| Carl Linnaeus | *Flora Zeylanica sistens plantas indicas Zeylonae insulae, quae olim 1670–77 lectae fuere a Paulo Hermanno* | 1747 | Latin | Fl. Zeyl. |
| Carl Linnaeus | *Skånska Resa* | 1751 | Swedish | It. scan., Iter Scan. |
| Carl Linnaeus | *Species Plantarum* | 1753 | Latin | Sp. pl., Spec. Plant. |
| Carl Linnaeus | *Iter Hispanicum, eller resa til Spanska Länderna uti Europa och America 1751 til 1756* | 1758 | Swedish | Loefl. It. |

A problem for Jörlin was to receive dyeing materials (*tinctoriae materiae*) from botanists, especially those from Asia and America (*Indiae alterutri*). Therefore, he searches in nature for local plants and collects all the parts that dyers need. Based on literature studies, he knows very well which dyestuffs were used in the other parts of the world.

The next topic is an introduction into the creation of colors. It is not Jörin's intention to explain the physics of color because the English physicist Isaac Newton (1642/43–1726/27) and others already did that. He mentions six primary colors (*colores primarios*): white (*album*), yellow (*luteum*), red (*rubrum*), blue (*caeruleum*), green (*viridem*) and black (*nigrum*). Color variations are obtained by mixing: grey (*cinereus*) from black and white; opal (*opalinus*) from white and blue; pink (*roseus*) from red and white; purple (*purpureus*) from blue and red; violet (*violaceus*) from blue and black; pale green (*pallide viridis*) from yellow and blue and dark green (*obscure viridis*) from blue and yellow.

He observed that some plants change color in the first stage of rotting, concluding that these plants could provide suitable dyes. The dried dog's mercury (*Mercurialis perennis* L.) and the yellow rattle (*Rhinanthus crista-galli* L., Rhinanthus minor L.) become grey and the flowers of *Lotus corniculatus* L. and *Anthyllis vulnerara* L. blue. Black discolorations are known from the black pea (*Orobus niger* L., current name: *Lathyrus niger* (L.) Bernh.), the gipsywort (*Lycopus europaeus* L.) and many roots of parasites, like *Hypocistis* (*Cytisus*), *Cynomorium*, *Monotropa* and *Lathraea* species.

In the case of gall and scale insects, Jörlin notes a close link between animals and plants. Galls are formed when insects lay their eggs in plants, like the *Quercus, Pistacia, Salix*, *Populus*, *Rosa*, *Glechoma*, *Hieracium* and *Statice* (*Limonium*) species. He names the host plants of the scale insects, like cacti (*Opuntia*) for American cochineal, oak (*Quercus*) for kermes, knawel (*Scleranthus perennis*) and bearberry (*Arctostaphylos uva-ursi*) for Polish cochineal.

White is created when naturally colored textile fibres of linen, wool and silk are properly rubbed and washed with soap. Although green is one of the most common colors in nature, it is rarely extracted from plants. According to Jörlin, a master dyer produces green colors by dyeing yellow and blue. Blue colors are hardly gained from indigenous plants. He states that the blue dyes occurring in the plant kingdom are wonderfully altered into a red color with acid and a green color with alkali (*colores caerulei, ab acido rubri, ab alcali virides evadunt*) with one exception, namely the stable indigo. The beautiful red color pleases the eyes with its brightness, but an even more intense color is created when acid is added. The dyers attribute the black color to the flora, which is not quite true because astringent (tannin-containing) plants need substances from the kingdom of stones, especially *martialia* (iron-containing agents). A strong and shiny blackness is achieved when the cloth is saturated with another color.

Linnaeus, the 'father of taxonomy' published his *Clavis Systematis Sexualis* in *Systema Naturae* in 1735 and the binomial nomenclature, the modern system for naming organisms, was for the first time applied in his *Species Plantarum* (1753). Jörlin names the plants and animals after this binomial nomenclature and classifies the dye plants according to the Linnaean system (Table 2). The plants are categorized on the basis of the numbers and characteristics of their male and female reproductive organs. The Rubiaceae have four male organs and are assigned to the *Tetrandria* (tetra = four, andr- = male). Weld (*Reseda luteola*) belongs to the *Dodecandria* (dodec- = twelve) and possesses twelve male organs. Woad (*Isatis tinctoria*) from the *Tetradynamia* (dynam- = power) has four long and two short male organs.

**Table 2.** Classification of the dyeing materials based on the Linnaean system.

| Linnaean System | Jörlin s Numbers | Linnaean System | Jörlin s Numbers | Linnaean System | Jörlin s Numbers |
| --- | --- | --- | --- | --- | --- |
| Monandria | 1 | Dodecandria | 43 | Gynandria | 75 |
| Diandria | 2–3 | Icosandria | 44–47 | Monoecia | 76–85 |
| Triandria | 4–7 | Polyandria | 48–54 | Dioecia | 86–89 |
| Tetrandria | 8–13 | Didynamia | 55–57 | Polygamia | 90 |
| Pentandria | 14–26 | Tetradynamia | 58 | Musci [1] | 91 |
| Hexandria | 27–30 | Diadelphia | 59–63 | Algae [2] | 92–97 |
| Octandria | 31–33 | Polyadelphia | 64 | Palmae | 98 |
| Decandria | 34–42 | Syngenesia | 65–75 | Animalia | 99–107 |

[1] In Jörlin's work, *Lycopodium complanatum* (Lycopodiaceae) is classified as a moss. [2] In Jörlin's work, lichens are classified as algae.

In addition to the dye plants, Jörlin also describes plants for washing textiles, namely *Struthium*, the common soapwort (*Saponaria officinals* L.) and two *Gypsophila* species of the family Caryophyllacae. He cites the Greek physician, pharmacologist and botanist Pedanius Dioscorides (ca. 40-ca. 90 CE) and the Roman politician and scholar Pliny the Elder (23/24-79 CE). Dioscorides says in *De materia medica* (50–70 CE) that fullers wash and clean cloths and wool with *Struthium* (*Saponaria officinals* L.) (Dsc. 2.163). Pliny the Elder in his *Naturalis Historia* (77 CE) notes that the plant rootlet (*radicula*), named *Struthion* by the Greek, provides a juice for washing and makes wool whiter and softer (NH 19.48). *Gypsophila struthium* L., native to Spain, is used by the Spaniards in the province of La Mancha to wash clothes and gives the same result as soap. Jörlin therefore recommends testing the washing properties of the native *Gypsophila fastigiata* L.

## 5. Yellows—*Lutei*

The majority of dyeing materials in Jörlin's dissertation concerns the yellows. 58 are listed and the most important fifteen species will be discussed first (Table 3). The herbs, leaves, flowers, fruits and old fustic possess flavonoid dyes, saffron contains the carotenoid dye crocetin, gamboge supplies xanthone dyes, barks provide mainly tannins and Indian sandalwood possesses unknown dyes. Seven of the 15 materials were imported: saffron, Avignon berries, tanner`s sumach and dyer's sumach from southern Europe or Asia, Indian sandalwood and gamboge from Asia and old fustic from Central and South America.

Jörlin notes that weld (*Reseda luteola*), abundantly imported from abroad, gives a beautiful chamois color and is favored by dyers. In a cloth factory in Norrköping, a city in the province of Östergötland, Linnaeus was told that weld had to be imported although it grew 'like a weed' everywhere in Lund, where it should be cultivated [19] (p. 10), [20] (p. 28). In the past, Sawwort (*Serratula tinctoria*), which dyes like weld, had to be imported, but in the mid-18th century the plant was so common that it was exported. Jörlin refers to Linnaeus, who wrote that double-dyeing for green with sawwort and indigo was practiced in the parish of Gothum on Gotland [19] (p. 224), [20] (p. 139). The plant grew wild around Uppsala, Stockholm, in Östergötland, Öland and Scania [19] (p. 10), [20] (p. 28). In the southern part of Sweden, the herb was gathered by poor people, mainly peasant women, sold to traders and exported to Copenhagen [21] (p. 156).

The bark of the alder buckthorn (*Frangula alnus*) as well as the bark of the common buckthorn (*Rhamnus cathartica*) provide beautiful yellow colors. Jörlin cites Linnaeus, who

states, that the inhabitants of Isgärde on Öland dye with the bark of the common buckthorn. In Martebo on Gotland, both barks are used and on Fårö, an island north of Gotland, these barks served to dye cloths [19] (pp. 58, 175, 209), [20] (p. 53, 115, 132). Persian berries belong to the Swedish imported goods, provide an excellent yellow, are highly esteemed and sold throughout Europe under the name of *grain d'Avignon*. Jörlin writes that these berries originate from the buckthorn species *Rhamnus minor* growing in southern Europe, especially in Spain and in the French commune Narbonne. The Linnaean Plant Name Typification Project website states that the precise synonymy of this name is uncertain and so the current name is *Rhamnus* sp.

The flowers of the dyer's broom (*Genista tinctoria*) are considered excellent for dyeing. The leaves of the moor birch (*Betula pubescens*), producing a faint yellow color, are used from time to time by farmers. Jörlin refers to the Finnish naturalist and chemist Pehr Adrian Gadd (1727–1797) who said that the leaves of the dwarf birch (*Betula nana*) provide a more excellent yellow dye than the moor birch [22] (p. 57). According to Linder, painters used the flowers of the dyer's broom to prepare the pigment *Schutgelb*, while the pigment *Sutgrön* was made from the leaves of the moor birch [23] (pp. 81–82). Dyer's broom, birch leaves, buckthorn berries and other flavonoid providing plants were suitable for the preparation of these pigments. Buckthorn berries were traded in Europe under the name Avignon berries (graines d'Avignon, French berries) and Persian berries (Aleppo berries, Smyrna berries) imported from the Levant. Depending on the ripeness of these berries, yellow and green pigments were prepared. Unripe berries were soaked in a lye (potash) and then precipitated with alum to create the transparent pretty yellow pigment *Schutgelb* (nowadays Stil de grain jaune, formerly pink or pinke). Stil de grain jaune was applied in medieval illuminated manuscripts, in 18th-century French and English paintings, in wallpapers and colored paper. *Sutgrön* (Stil de grain vert) was mostly made from ripe buckthorn berries precipitated with alum. The pigment was rarely used in painting. Both pigments have a poor lightfastness and fade rapidly [24].

The flowers of the dyer's chamomile (*Cota tinctoria*) give a bright yellow color and are highly valued among the native dyeing plants. Jörlin cites Linnaeus, who noted that in Gothum on Gotland the yarns are pre-mordanted with alum in a copper vessel and then boiled in a dye bath prepared with the dried flowers of dyer's chamomile, named *Johannis Blommor* (*Johannis Blomstor*) and *Flores Buphthalmi* [19] (pp. 223–224), [20] (p. 139). According to Linnaeus, *Chrysanthemum* flowers, similar to dyer's chamomile at first glance, were common in the southern Swedish provinces Scania and Halland. However, experiments showed that the *Chrysanthemum* flowers are not a proper substitute for dyer's chamomile.

Saffron, cultivated in southern Europe and Asia, consists of the stigmas of *Crocus sativus* and gives a very beautiful color, so it is widely known, not only for dyeing but also to color pharmaceuticals. The barks of the tanner's sumach (*Rhus coriaria*) and the dyer's sumach (*Cotinus coggygria*), both native to southern Europe and Asia, dye cloths. Jörlin claims that the *lignum medullaris* of Indian sandalwood (*Santalum album* L., Santalaceae) dyes yellow, while the exterior wood rarely occurs in the art of dyeing. The heartwood (*lignum medullaris*) has a darker yellow-to-brown color, while the outer sapwood is pale yellow-to-white [25] (p. 227, Figure 1). Gamboge, the thick yellow latex from *Garcinia hanburyi*, native to Indo-China, is more often used in painting than in dyeing. Old fustic, the wood of the dyer's mulberry (*Maclura tinctoria*) is an essential import commodity. The wood originates from a South American tree (*arbor Americae meridionalis*) and produces the most excellent color for dyeing yellow.

**Table 3.** Important native plants and trade products for yellow.

| Group | Jörlin-No. | Name in Jörlin / *Linnaean Name* | Material | English Name / *Swedish Name* | *Current Scientific Name* / Family |
|---|---|---|---|---|---|
| A | 19 / 19* | RHAMNUS frangula / *Rhamnus frangula* Linnaeus | Bark | Alder buckthorn / *Tröske* (*brakved*) | *Frangula alnus* Mill. / Rhamnaceae |
| A | 20 / 20* | RHAMNUS catharticus / *Rhamnus cathartica* Linnaeus | Bark | Common buckthorn / *Getapel* | *Rhamnus cathartica* L. / Rhamnaceae |
| A | 43 / 41* | RESEDA Luteola / *Reseda luteola* Linnaeus | Herb | Weld / *Wau* | *Reseda luteola* L. / Resedaceae |
| A | 62 / 62* | GENISTA tinctoria / *Genista tinctoria* Linnaeus | Flowers | Dyer s broom / *Ginst,* (*färgginst*) | *Genista tinctoria* L. / Fabaceae |
| A | 66 / 66* | SERRATULA tinctoria / *Serratula tinctoria* Linnaeus | Herb | Sawwort / *Ängskara* | *Serratula tinctoria* L. / Asteraceae |
| A | 70 / 70* | ANTHEMIS tinctoria / *Anthemis tinctoria* Linnaeus var. tinctoria | Flowers [1] | Dyer s chamomile / *Lettblomster,* (*färgkulla, Johannis Blomster*) | *Cota tinctoria* (L.) J.Gay / Asteraceae |
| A | 76 / 76* | BETULA alba / *Betula alba* Linnaeus | Leaves | Moor birch / *Biork, björk* | *Betula pubescens* Ehrh. / Betulaceae |
| A | 77 / 77* | BETULA nana / *Betula nana* Linnaeus | Leaves | Dwarf birch / *Skarre,* (*dvärgbjörk*) | *Betula nana* L. / Betulaceae |
| B | 5 / - | CROCUS sativus [2] / *Crocus sativus* Linnaeus | Stigmas | Saffron / *Safran* | *Crocus sativus* L. / Iridaceae |
| B | 21 / 21* | RHAMNUS minor / *Rhamnus minor* Linnaeus | Fruits | Avignon berry / *Grain d Avignon* | *Rhamnus* sp. / Rhamnaceae |
| B | 24 / 24* | RHUS coriaria / *Rhus coriaria* Linnaeus | Bark | Tanner s sumach / *Sumach,* (*bärsumak*) | *Rhus coriaria* L. / Anacardiaceae |
| B | 25 / - | RHUS Cotinus / *Rhus cotinus* Linnaeus | Bark | Smoke tree, dyer s sumach / (*Perukbuske*) | *Cotinus coggygria* Scop. / Anacardiaceae |
| B | 31 / 31* | SANTALUM album / *Santalum album* Linnaeus | Wood | Indian sandalwood / *Sandel* | *Santalum album* L. / Santalaceae |
| B | 49 / 49* | CAMBOGIA Gutta / - | Latex | Gamboge / *Gummi-gutta* | *Garcinia hanburyi* Hook.f. and other *Garcinia* species / Clusiaceae |
| B | 79 / 79* | MORUS tinctoria / *Morus tinctoria* Linnaeus | Wood | Old fustic, dyer s mulberry / (*Fustikträdet*) | *Maclura tinctoria* (L.) D.Don ex Steud. / Moraceae |

Swedish names between brackets are from other sources than Jörlin; * Number occurring in the index sorted by color in Jörlin [3] (p. 342); [1] *Flores Buphthalmi*; [2] Jörlin places saffron in the category of reds (*rutili*); A = native plant, used in Sweden; B = imported trade product.

Eighteen materials are of less importance for dyeing yellow in Sweden (Table 4). The herbs, leaves and flowers provide flavonoid dyes, bark and roots of the common barberry yield the alkaloid dye berberine, and other barks contain mainly tannins.

The herb of the three-lobe beggartick (*Bidens tripartita*) provides a yellow dye. It was known in Scania for yellow and brown, while yellow and orange colors were produced near Isgärde, Hulterstad and Sandby on Öland [19] (pp. 58, 98, 101), [20] (pp. 53, 73, 75), [26] (pp. 240, 277). Rather seldom peasants take the herb of the woundwort (*Anthyllis vulneraria*) to dye their clothes. In Scania, five plants are rarely used by farmers: the dried leaves of devil's bit (*Succisa pratensis*), the herb of Kalm's hawkweed (*Hieracium umbellatum*), the herb of bog-myrtle (*Myrica gale*), the dried leaves of the bay willow (*Salix pentandra*) and the dried leaves of the purple willow (*Salix purpurea*) [26] (pp. 277, 293, 342). The ground-cedar (*Lycopodium complanatum*) dyes a very beautiful yellow and is often applied in rural areas. It should be noted that this dye plant accumulates the element aluminum in the cell sap and can serve as a mordant. In the dye works of Växjö in Småland, the brown knapweed (*Centaurea jacea*) is used instead of sawwort (*Serratula tinctoria*), but experiments show that the brown knapweed dyes less perfectly than sawwort [19] (p. 307), [20] (p. 183).

The corollas of the common marigold (*Calendula officinalis*) yield a yellow dye after squeezing and boiling with alum, and sometimes farmers take the dried corollas instead of saffron. According to the Finnish botanist and naturalist Pehr Kalm (1716–1779), the inflorescences of lady's bedstraw (*Galium verum*), boiled with alum, dye woolen clothes yellow [27] (pp. 251–252), but the roots on the other hand dye red (chapter Reds—rutili). The flowers of St John's wort (*Hypericum perforatum*) taken by rural people are of lesser value. The yellow bark of the apple tree (*Malus* sp.) is used by dyers, the bark of the plum tree (*Prunus domestica*) by country dwellers and the yellow bark of the common hornbeam (*Carpinus betulus*) mainly by Scanians. In Poland, a very pale saffian color is obtained on leather with the yellow bark of the common barberry (*Berberis vulgaris*), and in Sweden its roots are soaked in lye to dye wool. By boiling eggs with the yellow roots of the common nettle (*Urtica dioica*), peasants color the eggshells, especially at Easter.

**Table 4.** Less important native plants used for yellow.

| Grou | Jörlin -No. | Name in Jörlin<br>*Linnaean Name* | Material | **English Name**<br>*Swedish Name* | *Current Scientific Name*<br>Family |
|---|---|---|---|---|---|
| A | 8<br>8* | SCABIOSA Succisa<br>*Scabiosa succisa* Linnaeus | Leaves | Devil s-bit<br>*Angwadd* | *Succisa pratensis* Moench<br>Dipsacaceae |
| A | 11<br>- | GALIUM verum<br>*Galium verum* Linnaeus | Flowers | Lady s bedstraw<br>*Mariae Sanghalm* | *Galium verum* L.<br>Rubiaceae |
| A | 27<br>27* | BERBERIS vulgaris<br>*Berberis vulgaris* Linnaeus | Bark | Common barberry<br>*Berberis* | *Berberis vulgaris* L.<br>Berberidaceae |
| A | 27<br>27* | BERBERIS vulgaris<br>*Berberis vulgaris* Linnaeus | Roots | Common barberry<br>*Berberis* | Berberis vulgaris L.<br>Berberidaceae |
| A | 44<br>42* | PRUNUS domestica<br>*Prunus domestica* Linnaeus | Bark | European plum<br>*Plommonträd* | *Prunus domestica* L.<br>Rosaceae |
| A | 45<br>43* | PYRUS Malus<br>*Pyrus malus* Linnaeus | Bark | Apple tree<br>*Appel* | *Malus* sp.<br>Rosaceae |
| A | 61<br>61* | ANTHYLLIS Vulneraria<br>*Anthyllis vulneraria* Linnaeus | Herb | Kidneyvetch, wound-wort<br>*Rafklor (getväppling)* | *Anthyllis vulneraria* L.<br>Fabaceae |
| A | 64<br>64* | HYPERICUM perforatum<br>*Hypericum perforatum* Linnaeus | Flowers | St. John s wort<br>*Johannisort* | *Hypericum perforatum* L.<br>Hypericaceae |

| A | 65<br>65* | HIERACIUM umbellatum<br>*Hieracium umbellatum* Linnaeus | Herb | Kalm s hawkweed<br>*(Flockfibbla)* | *Hieracium umbellatum* L.<br>Asteraceae |
|---|---|---|---|---|---|
| A | 68<br>38* | BIDENS tripartite<br>*Bidens tripartita* Linnaeus | Herb | Three-lobe beggartick<br>*Brunskär, (brunskära)* | *Bidens tripartita* L.<br>Asteraceae |
| A | 71<br>71* | CENTAUREA Jacea<br>*Centaurea jacea* Linnaeus | Herb | Brown knapweed<br>*Knappar, (rödklint)* | *Centaurea jacea* L.<br>Asteraceae |
| A | 73<br>73* | CALENDULA officinalis<br>*Calendula officinalis* Linnaeus | Flowers | Common marigold<br>*Ringblomma* | *Calendula officinalis* L.<br>Asteraceae |
| A | 80<br>80* | URTICA dioica<br>*Urtica dioica* Linnaeus | Roots | Common nettle<br>*Näßla, (brännässla)* | *Urtica dioica* L.<br>Urticaceae |
| A | 84<br>84* | CARPINUS Betulus<br>*Carpinus betulus* Linnaeus | Bark | Common hornbeam<br>*Afwenbok, (avenbok)* | *Carpinus betulus* L.<br>Fagaceae |
| A | 86<br>86* | SALIX pentandra<br>*Salix pentandra* Linnaeus | Leaves | Bay willow<br>*Jolster* | *Salix pentandra* L.<br>Salicaceae |
| A | 86<br>86* | SALIX purpurea<br>*Salix purpurea* Linnaeus | Leave | Purple willow<br>*(Rödvide)* | *Salix purpurea* L.<br>Salicaceae |
| A | 88<br>88* | MYRICA gale<br>*Myrica gale* Linnaeus | Herb | Bog-myrtle<br>*Pors* | *Myrica gale* L.<br>Myricaceae |
| A | 91<br>91* | LYCOPODIUM complanatum<br>*Lycopodium complanatum* Linnaeus | Herb | Ground-cedar<br>*Jemna* | *Lycopodium complanatum* L.<br>Lycopodiacae |

Swedish names between brackets are from other sources than Jörlin; * Number occurring in the index sorted by color in Jörlin [3] (p. 342); A = native plant, used in Sweden.

Jörlin lists sixteen materials from native plants with the potential for dyeing yellow and five well-known dyestuffs used abroad (Table 5). Nearly all contain flavonoid dyes, turmeric the yellow dye curcumin and galls mainly tannins.

*Persicaria maculosa* and *Persicaria hydropiper* dye woolen cloths, mordanted with alum, the herb of yellow loosestrife (*Lysimachia vulgaris*) as well as the roots and leaves of the common meadow-rue (*Thalictrum flavum*) dye wool, and an ink can be prepared from the flowers of marsh-marigold (*Caltha palustris*). Furthermore, yellows are created with the herb of yellow loosestrife (*Lysimachia vulgaris*), the herb of hedge woundwort (*Stachys sylvatica*), herb and fruits of the rough cocklebur (*Xanthium strumarium*), the flower heads (umbels) of cow parsley (*Anthriscus sylvestris*) and galls occurring on willows (*Salix* sp.).

Jörlin observes that the green parts of the seashore dock (*Rumex maritimus*), the false hemp (*Datisca cannabina*) and the staghorn sumac (*Rhus typhina*) turn yellow during ripening in the summer. He concludes that they provide yellow dyes and suggests experiments with these plants.

Jörlin describes various plants from abroad. The roots (actually rhizomes) of two plants, namely turmeric (*Curcuma longa*) and the Chinese keys (*Boesenbergia rotunda*), both native to India, dye a rich but fugitive color. The Spaniards dye yellow with the flower heads (umbels) of the villous deadly carrot (*Thapsia villosa*). The English vernacular name points to the poison of the roots, which was applied by fishermen in Catalonia as ichthyotoxin to stun fish to make them easier to catch [28]. Referring to the French traveller and naturalist Pierre Belon (Petrus Bellonius, 1517–1564), Jörlin writes that the pistachio galls serve for the preparation of yellows in the Orient. Jörlin mentions that the herb of bear's breeches (*Acanthus mollis*) has been known since ancient times, citing the epic Aeneid

(1:653) of the Roman poet Publius Vergilius Maro, named Virgil (70–19 BCE): "And the veil woven around (with a border) with saffron-yellow acanthus" (*Et circumtextum croceo velamen Acantho*). Another explanation is that the border of the veil possessed a pattern of acanthus leaves as it occurs in Late Antique textiles.

**Table 5.** Native plants with potential use for yellow and non-native plants used abroad.

| Grou | Jörlin -No. | Name in Jörlin<br>*Linnaean Name* | Material | English Name<br>*Swedish Name* | *Current Scientific Name*<br>Family |
|------|------|------|------|------|------|
| C | 16<br>16* | LYSIMACHIA vulgaris<br>*Lysimachia vulgaris* Linnaeus | Herb | Yellow loosestrife<br>(*Strandlysing*) | *Lysimachia vulgaris* L.<br>Primulaceae |
| C | 22<br>22* | CHAEROPHYLLUM sylvestre<br>*Chaerophyllum sylvestre* Linnaeus | Flowers | Cow parsley<br>*Hundkaxa* | *Anthriscus sylvestris* (L.) Hoffm.<br>Apiaceae |
| C | 30<br>30* | RUMEX maritimus [1]<br>*Rumex maritimus* Linnaeus | Herb | Seashore dock<br>*Hafssyra* | *Rumex maritimus* L.<br>Polygonaceae |
| C | 33<br>33* | POLYGONUM persicaria<br>*Polygonum persicaria* Linnaeus | Herb | *Jungfru-twäl* | *Persicaria maculosa* Gray<br>Polygonaceae |
| C | 33<br>33* | POLYGONUM hydropiper<br>*Polygonum hydropiper* Linnaeus | Herb | *Jungfru-twäl* | *Persicaria hydropiper* (L.) Delarbre<br>Polygonaceae |
| C | 53<br>53* | THALICTRUM flavum<br>*Thalictrum flavum* Linnaeus | Roots | Common meadow-rue; (*Ängsruta*) | *Thalictrum flavum* L.<br>Ranunculaceae |
| C | 53<br>53* | THALICTRUM flavum<br>*Thalictrum flavum* Linnaeus | Leaves | Common meadow-rue; (*Ängsruta*) | *Thalictrum flavum* L.<br>Ranunculaceae |
| C | 54<br>54* | CALTHA palustris<br>*Caltha palustris* Linnaeus | Flowers | Marsh-marigold<br>*Kabbelek* | *Caltha palustris* L.<br>Ranunculaceae |
| C | 56<br>56* | STACHYS sylvatica<br>*Stachys sylvatica* Linnaeus | Herb | Hedge woundwort<br>(*Stinksyska*) | *Stachys sylvatica* L.<br>Lamiaceae |
| C | 81<br>81* | XANTHIUM strumarium<br>*Xanthium strumarium* Linnaeus | Herb | Rough cocklebur, clotbur<br>(*Gullfrö, ljust gullfrö*) | *Xanthium strumarium* L.<br>Asteraceae |
| C | 81<br>81* | XANTHIUM strumarium<br>*Xanthium strumarium* Linnaeus | Fruits | Rough cocklebur, clotbur<br>(*Gullfrö, ljust gullfrö*) | *Xanthium strumarium* L.<br>Asteraceae |
| C | 89<br>89* | DATISCA cannabina<br>*Datisca cannabina* Linnaeus | Herb | False hemp, acalbir<br>- | *Datisca cannabina* L.<br>Datiscaceae |
| C | 89<br>89* | DATISCA hirta<br>*Datisca hirta* Linnaeus | Herb | Staghorn sumac<br>(*Rönnsumak*) | *Rhus typhina* L.<br>Anacardiaceae |

| | | | | | |
|---|---|---|---|---|---|
| C | 99 - | CYNIPS Salicis viminalis - | Galls | Galls - | *Euura viminalis* Kopelke, 2001; *Cynips salicis* subsp. viminalis Christ, 1791 Tenthredinidae |
| C | 99 - | CYNIPS Salicis amerinae *Cynips amerinae* Linnaeus, 1758 | Galls | Galls on bay willow (*Salix pentandra* L.) | *Euura amerinae* (Linnaeus, 1758) Tenthredinida |
| C | 99 - | CYNIPS Salicis strobili *Cynips salicisstrobili* Linnaeus, 1758 | Galls | Galls - | *Pseudencyrtus salicisstrobili* (Linnaeus, 1758) Encyrtidae |
| D | 1 1* | CURCUMA longa *Curcuma longa* Linnaeus | Roots [2] | Turmeric *Garkmäja* | *Curcuma longa* L. Zingiberaceae |
| D | 1 1* | CURCUMA rotunda *Curcuma rotunda* Linnaeus | Roots [2] | Chinese keys *Garkmäja* | *Boesenbergia rotunda* (L.) Mansf. Zingiberaceae |
| D | 23 - | THAPSIA villosa *Thapsia villosa* Linnaeus | Flowers | Villous deadly carrot - | *Thapsia villosa* L. Apiaceae |
| D | 57 57* | ACANTHUS mollis *Acanthus mollis* Linnaeus | Herb | Bear s breeches (*Mjukakantus*) | *Acanthus mollis* L. Acanthaceae |
| D | 101 108* | APHIDES Pistaciae [3] *Aphis pistaciae* Linnaeus 1767 | Galls | Pistachio galls - | *Baizongia pistaciae* (Linnaeus, 1767) on Pistacia terebinthus L.; *Aploneura lentisci* (Passerini 1856) on *Pistacia lentiscus* L. Aphididae |

Swedish names between brackets are from other sources than Jörlin; * Number occurring in the index sorted by color in Jörlin [3] (p. 342); [1] In the index, Jörlin lists the roots and not the herb; [2] actually rhizomes; [3] Jörin mentions these galls for dyeing yellow and red, but in the index, it is only placed for dyeing red (*rutili*); C = native plant with potential use for dyeing; D = non-native plant, used abroad.

## 6. Golden Reds—*Rutili*

For dyeing golden red, three materials are listed (Table 6). *Alkanna* (henna) contains the naphthquinone Lawson, safflower the red dye carthamin and pistachio galls mainly tannins.

Jörlin mentions *Lawsonia inermis* as *Alkanna*, a plant native to Asia and Africa, mainly cultivated in Egypt. It should be noted that the name *Alkanna* dates back to the Arabic term *al-ḥinnā*, meaning henna. Today, *Alkanna* is the genus name for alkanet, which Jörlin names *Pseudo Alcanna* (Table 8). He thinks that the dried herb of *Lawsonia inermis* applied by the Orientals yields an excellent yellow to dye parts of their bodies and further he believes that the roots treated with quicklime (*calx viva*, calcium oxide) yield a red dye to color teeth, nails, faces, manes of horses, hides, wood, waxes, ointments, decoctions and cloths. Today, it is known that the dye originates from the leaves, but not from the herb and roots. According to Jörlin, the corolla of safflower (*Carthamus tinctorius*) treated with acid provides a very bright pink color, which is especially charming on silk (*Corollae Roseum colorem acido praeparatae praebent nitidissimum, serica inprimis venuste colorantem*). For this reason, he proposes the cultivation of safflower, as Linnaeus recommended for southern Sweden twenty years earlier [19] (p. 34), [20] (p. 40). Jörlin quotes Belon, who wrote that the Orientals took pistachio galls to prepare red colors using acid.

Table 6. Non-native plants for golden reds.

| Group | Jörlin-No. | Name in Jörlin<br>*Linnaean Name* | Material | **English Name**<br>*Swedish Name* | *Current Scientific Name*<br>Family |
|---|---|---|---|---|---|
| D | 32<br>32* | LAWSONIA inermis<br>*Lawsonia inermis* Linnaeus | Leaves | Henna tree, henna<br>*Alkanna* | *Lawsonia inermis* L.<br>Lythraceae |
| D | 67<br>- | CARTHAMUS tinctorius<br>*Carthamus tinctorius*<br>Linnaeus | Flowers | Safflower<br>*Safflor* | *Carthamus tinctorius* L.<br>Asteraceae |
| D | 101<br>108* | APHIDES Pistaciae<br>*Aphis pistaciae*<br>Linnaeus 1767 | Galls | Pistachio galls<br>- | *Baizongia pistaciae* (Linnaeus, 1767)<br>on *Pistacia terebinthus* L.;<br>*Aploneura lentisci* (Passerini 1856)<br>on *Pistacia lentiscus* L.<br>Aphididae |

* Number occurring in the index sorted by color in Jörlin [3] (p. 342); D = non-native plant, used abroad.

## 7. Reds—*Rubri*

For dyeing red, 24 materials are listed, which are divided in two groups (Tables 7 and 8). The first group includes the roots (actually rhizomes) of four Rubiaceae species and five scale insects, which all provide anthraquinone dyes (Table 7). The following four materials were imported: dyer's madder and kermes from the Mediterranean region, lac dye from Asia and American cochineal from Central and South America.

Jörlin reports that the imported dyer's madder (*Rubia tinctorum*) as well as indigenous Rubiaceae species are commonly used by the dyers. On the Baltic islands, the roots of dyer's woodruff (*Asperula tinctoria*) are collected before the stems sprout. It is proved by dyers' experiments that these roots provide an excellent substitute for dyer's madder. He refers to Linnaeus: "It is essential that the roots be collected before the cuckoo starts calling, that is before the roots put up stems; since the roots are looser then and yield more color... One cooks them with the sourest beer, especially the kind of *Standebilla*, for the sourer the beer is the more intense the color becomes… After boiling the roots, the yarns or stockings are put into the decoction, while it is still warm, then they are rinsed quickly in a weak lye" [19] (pp. 238–239), [20] (p. 147). The malt left over after the beer production was treated again with water to produce a very sour malt brew, which the Gotlanders gave in their dye baths because only limewater occurs on Gotland [29] (pp. 20–21). *Asperula tinctoria* was found near Päsnäs on Öland, on Stora Karlsö, an island off the west coast of Gotland and in the vicinities of Visby, Martebo, Rute, När and Alskog on Gotland, where it grew in such quantities that it could be collected for dye works [19] (115–116, 168, 187, 195, 238–239, 283), [20] (pp. 84, 111, 120, 125, 147, 171). Jörlin reports that the farmers in Finland take the roots of the northern bedstraw (*Galium boreale*) for dyeing wool. He cites Kalm who stated that the roots of lady's bedstraw (*Galium verum*) also give a red color [27] (p. 244).

Other anthraquinone sources are the female scale insects of American cochineal (*Dactylopius coccus*), kermes (*Kermes ilicis*), Polish cochineal (*Porphyrophora polonica*) and lac dye (*Kerria lacca*). In the tropical greenhouse (*Caldarium*) of the Botanical (Linnaean) Garden in Uppsala, Jörlin saw cochineal insects (*Dactylopius coccus*) vividly some years ago and writes enthusiastically about this dye: "Dried pregnant females give such a nice red color, that today we can easily live without the purple of our ancestors, and there is nothing more frequent in the art of dyeing".

He states that dyers and pharmacists use the insects of *Kermes ilicis*. It can be assumed that Jörlin means kermes (*Kermes vermilio* Planchon, 1864) because *Kermes ilicis* provides

only "slightly pinkish beige-browns" [8] (p. 609). Jörlin considers *Kermes ilicis*, American cochineal (*Dactylopius coccus*) and redwood (*Caesalpinia*) as organic coloring matters precipitated on alum for a Florentine Lake. Jörlin claims that Polish cochineal (*Porphyrophora polonica*) is native to Sweden where he discovered the species in the summer of 1858. This dye insect is found on the roots of three plants, the perennial knawel (*Scleranthus perennis* L.), the mouse-ear hawkweed (*Hieracium pilosella* L.) and the bearberry (*Arctostaphylos uva-ursi* (L.) Spreng.). In his view, the color of Polish cochineal is not inferior to American cochineal. Regarding the oystershell scale (*Lepidosaphes ulmi*), he only says that it is a native insect that provides red. Jörlin describes lac insects (*Insecta laccifera*), referring to the German physician and botanist Paul Hermann (Hermannus, 1646–1695) who was a medical officer to the Dutch East India Company in Ceylon between 1670 and 1677, where he collected many plants, particularly in the area around Colombo. Hermannus stated that these insects, native to *Indiae orientalis*, provide a resinous secretion on *Croton aromaticus* L. (*Croton laccifer* L.), sold as lacca. According to Jörlin, lacca is often used for dyeing pharmaceuticals and clothlets (*bezetta*).

Clothlets were prepared according to a historic method by soaking small pieces of tissue in a juice or liquid mainly derived from plants, which were then dried in order to store a water-soluble coloring matter until needed. They were made from the finest crepon, cotton and Dutch linen especially in red hues but also in other colors, like pale or dark yellow, green, blue, brown and violet. The most expensive colors were rose and carmoisin. Dyes for red and violet clothlets were obtained from American cochineal (*Dactylopius coccus*) and dyer's croton (tournesol, *Chrozophora tinctoria*), but Jörlin knows that lac dye (*Kerria lacca*), dyer's alkanet (*Alkanna matthioli*) and inkberries (*Phytolacca americana*) served for red *bezetta* and Malabar spinach (*Basella alba*) for violet *bezetta*. When artists needed them as coloring agents, these tissues were moistened in a small container with a little water. This technique has been known since the Middle Ages, mainly in manuscript illumination and later in watercolor. The technique of making pezzette was already described by the Italian painter Cennino Cennini (c. 1360-before 1427) in his 1390 *Il libro dell'arte*. Clothlets also served in cosmetics for make-up and to color wax, liqueurs, confectionary, baked goods, confitures, jellies and creams [30].

**Table 7.** Rubiaceae species and scale insects for red.

| Group | Jörlin-No. | Name in Jörlin *Linnaean Name* | Material | English Name *Swedish Name* | *Current Scientific Name* **Family** |
|---|---|---|---|---|---|
| A | 9 9* | ASPERULA tinctoria *Asperula tinctoria* Linnaeus | Roots [1] | Dyer s woodruff *Madra* | *Asperula tinctoria* L. Rubiaceae |
| A | 10 10* | GALIUM boreale *Galium boreale* Linnaeus | Roots [1] | Northern bedstraw *Mära* | *Galium boreale* L. Rubiaceae |
| A | 11 11* | GALIUM verum *Galium verum* Linnaeus | Roots [1] | Lady s bedstraw *Mariae Sanghalm* | *Galium verum* L. Rubiaceae |
| B | 12 12* | RUBIA tinctorum *Rubia tinctorum* Linnaeus | Roots [1] | Dyer s madder *Krapp* | *Rubia tinctorum* L. Rubiaceae |
| B | 102 102* | INSECTA laccifera - | Insects | Lac dye *Lacca* | *Kerria lacca* (Kerr, 1782) Kerriidae |
| B | 103 103* | COCCUS ilicis *Coccus ilicis* Linnaeus, 1758 | Insects | - - | *Kermes ilicis* (Linnaeus, 1758) [2] Kermesidae |
| B | 104 | COCCUS cacti | Insects | American cochineal | *Dactylopius coccus* Costa, 1829 |

| | | | | | |
|---|---|---|---|---|---|
| | 104* | Coccus cacti Linnaeus, 1758 | | *Coccionell* | Dactylopiidae |
| C | 105 105* | COCCUS Polonicus *Coccus polonicus* Linnaeus, 1758 | Insects | Polish cochineal - | *Porphyrophora polonica* (Linnaeus, 1758) Margarodidae |
| C | 106 106* | COCCUS Ulmi *Coccus ulmi* Linnaeus, 1758 | Insects | Oystershell scale - | *Lepidosaphes ulmi* (Linnaeus, 1758) Diaspididae |

\* Number occurring in the index sorted by color in Jörlin [3] (p. 342); [1] actually rhizomes; [2] Meant is kermes, *Kermes vermilio* Planchon, 1864; A = native plant, used in Sweden; B = imported trade product; C = native animal with potential use for dyeing.

The second group of reds contains eight materials with different red dyes and seven tannin-rich plants (Table 8). The roots of two Boraginaceae species contain the naphthoquinone dye alkannin, the ripe buckthorn berries and the European dodder anthocyanins, the red dye nordin occurs in dragon's blood, the homoisoflavonoid dye brazilein in brazilwood, the carotinoid bixin in annatto and the betacyanin dye phytolaccanin in inkberries. Tormentil roots, the root bark of tanner's sumach, the root bark of dyer's sumach, alder bark, common sorrel, native sorrel and catechu mainly provide tannins. The following five materials were imported: dyer's alkanet from the Mediterranean region, dragon's blood from India, the fruits the Areca palm from Asia, brazilwood and annatto from Central and South America.

Jörlin mentions that the roots of imported dyer's alkanet (*Alkanna matthioli*) give a color, which is not lightfast and is hardly applicable for dyeing pharmaceuticals except for *bezetta*. He explains that the roots of the common bugloss (*Anchusa officinalis*) can replace the roots of dyer's alkanet. Linnaeus reported that another Boraginaceae species served as a kind of make-up, particularly in Hälsingland, a province in central Sweden belonging to Norrland [31] (p. 46). Young women took the fresh roots of the field gromwell (*Buglossoides arvensis* (L.) I.M.Johnst., *Lithospermum arvense* L.), washed them slightly and colored their faces with them. It gave them a pleasant red color on their cheeks that was attractive to suitors [21] (pp. 158–159).

The herb of the European dodder (*Cuscuta europaea*) yields a pale reddish shade, and the ripe buckthorn berries (*Rhamnus cathartica*) provide a pink juice when picked in the late fall. The German naturalist, physician and explorer Engelbert Kaempfer (Kaempferus, 1651–1716) reported that the fruits of *Calamus rotang*, native to India, are a source of dragon's blood, but Jörlin assumes that also other genera provide this resinous substance. He further notes its application by doctors to color pharmaceuticals. Regarding imported brazilwood (*Caesalpinia brasiliensis*), he only says that it is very common in dyeing. Annatto from the American pokeweed (*Bixa orellana*) is used by dyers and the American Indians paint their naked bodies with these seeds. The juice of the inkberries (*Phytolacca americana*) gives a fugitive pink dye for tinctures and *bezetta*.

The tormentil roots (*Potentilla erecta*) provide a red color and Linnaeus reported that chewed roots were smeared on leather in Lapland [31] (p. 171, No. 213). The alder bark (*Alnus* sp.), the roots of the purple marshlocks (*Potentilla palustris*), the root barks of the tanner's sumach (*Rhus coriaria*) and the root barks of the dyer's sumach (*Cotinus coggygria*) are well suited for dyeing red. The dried roots of the native sorrel (*Rumex acetosa*) are distilled for pharmaceuticals but not for dyeing textiles. The unripe fruits of the betel palm (*Areca catechu*), growing in tropical Asia, are boiled in water with quicklime (*calx viva*, calcium oxide) to produce the 'catechu of pharmacists' that serves more for coloring medicine than for dyeing.

**Table 8.** Other dyeing materials for red.

| Group | Jörlin -No. | Name in Jörlin / *Linnaean Name* | Material | English Name / *Swedish Name* | *Current Scientific Name* / Family |
|---|---|---|---|---|---|
| A | 29 / 29* | RUMEX Acetosa / *Rumex acetosa* Linnaeus | Roots | Common sorrel / *Syra, (ängssyra)* | *Rumex acetosa* L. / Polygonaceae |
| A | 46 / 46* | TORMENTILLA erecta / *Tormentilla erecta* Linnaeus | Roots | Tormentil / *Blodrot* | *Potentilla erecta* (L.) Raeusch. / Rosaceae |
| A | 78 / - | BETULA Alnus / *Betula alnus* Linnaeus | Bark | Black alder and grey alder / *Al* | *Alnus glutinosa* (L.) Gaertn. and *Alnus incana* (L.) Moench; Betulaceae |
| B | 15 / 15* | LITHOSPERMUM tinctorium / *Lithospermum tinctorium* Linnaeus | Roots | Dyer s alkanet / *Pseudo Alcanna* | *Alkanna matthioli* Tausch, syn. *Alkanna tinctoria* Tausch subsp. *tinctoria* / Boraginaceae |
| B | 28 / 28* | CALAMUS Rotang / *Calamus rotang* Linnaeus | Resin | Dragon s blood / *Sangvis draconis* | *Calamus rotang* L. / Arecaceae |
| B | 35 / 35* | CAESALPINIA brasiliensis / *Caesalpinia brasiliensis* Linnaeus | Wood | Brazilwood / *Brasilja* | *Caesalpinia brasiliensis* L. / Fabaceae, subfamily Caesalpinioideae |
| B | 50 / 50* | BIXA Orellana / *Bixa orellana* Linnaeus | Seeds | Annatto / *Orlean* | *Bixa orellana* L. / Bixaceae |
| B | 98 / 98* | ARECA Cathecu / *Areca catechu* Linnaeus | Fruits | Areca palm / *Terra Cathecu* | *Areca catechu* L. / Arecaceae |
| C | 13 / 13* | CUSCUTA europaea / *Cuscuta europaea* Linnaeus | Herb | European dodder / *Snar-refwa* | *Cuscuta europaea* L. / Convolvulaceae |
| C | 14,15 / - | ANCHUSA officinalis / *Anchusa officinalis* Linnaeus | Roots | Common bugloss / *Oxtunga* | *Anchusa officinalis* L. / Boraginaceae |
| C | 20 / - | RHAMNUS catharticus / *Rhamnus cathartica* Linnaeus | Berries | Common buckthorn / *Getapel* | *Rhamnus cathartica* L. / Rhamnaceae |
| C | 47 / 47* | COMARUM palustre / *Comarum palustre* Linnaeus | Roots | Purple marshlocks / *Krakfotter, (kråkklöver)* | *Potentilla palustris* (L.) Scop. / Rosaceae |
| D | 24 / - | RHUS coriaria / *Rhus coriaria* Linnaeus | Roots | Tanner s sumach / *Sumach, (bärsumak)* | *Rhus coriaria* L. / Anacardiaceae |
| D | 25 / 25* | RHUS Cotinus / *Rhus cotinus* Linnaeus | Roots | Smoke tree, dyer s sumach / *(Perukbuske)* | *Cotinus coggygria* Scop. / Anacardiaceae |
| D | 42 / 42* | PHYTOLACCA americana / *Phytolacca americana* Linnaeus | Berries | American pokeweed, inkberry / *(Amerikanskt kermesbär)* | *Phytolacca americana* L. / Phytolaccaceae |

Swedish names between brackets are from other sources than Jörlin; * Number occurring in the index sorted by color in Jörlin [3] (p. 342); A = native plant, used in Sweden; B = imported trade product; C = native plant with potential use for dyeing; D = non-native plant used abroad.

### 8. Blues—*Caerulei*

For dyeing blue, ten materials are listed (Table 9). Four indigo plants provide the blue pigment indigotin, blue flowers of three species contain anthocyanins, yellow flowers of two species seem to provide unknown blue dyes, tournesol yields chrozophoridin and the ash bark an unknown blue dye. The following three materials were imported: woad from the neighboring countries, indigo from India and dyer's croton from the Mediterranean region or Asia.

Jörlin writes on woad (*Isatis tinctoria*) that it is planted in Sweden, although more than a fair amount is imported from neighboring countries. Further, he points to a woad mill at Alingsås in Västergötaland as indicated by Linnaeus (Figure 5) [32] (pp. 127–128). For the processing of woad, Jörlin quotes the *Historia Plantarum* by the British botanist John Ray (1627–1705) [33] (p. 842). Woad was mixed with indigo in a *blåkyppen* (blue vat). Linnaeus observed that woad grows wild near Torp in the north of the island Öland and recommended its cultivation on Lillholmen, a small island near Stockholm, and on the island of Klasen north of Gotland, where he found the plant on the coast [19] (pp. 111–112, 144, 216), [20] (pp. 82, 98, 135).

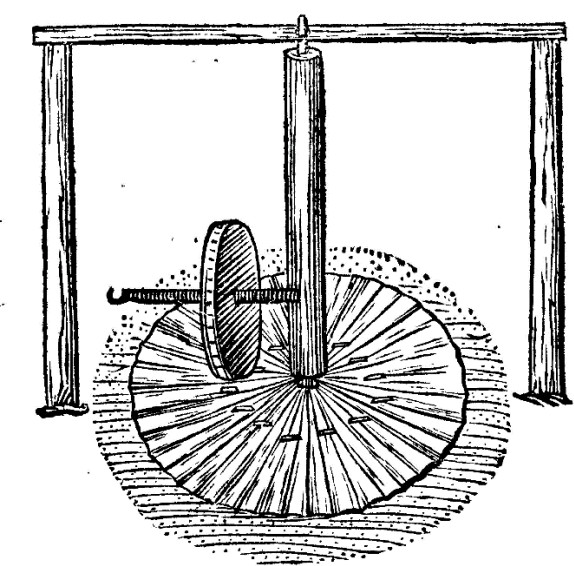

**Figure 5.** Woad mill in Alingsås in Västergotaland [32] (p. 128). Image: Niedersächsische Staats- und Universitätsbibliothek Göttingen, Germany, Signatur 8 H NAT III, 1380.

Indigo gained from *Indigofera tinctoria*, native to India and Indo-China, is the most excellent among all known dyestuffs and is therefore very popular. It has to be imported at a high price from foreigners, so efforts are still made to find a European plant as a substitute. Jörlin cites Ray's *Historia Plantarum*, which described the production of indigo as observed by the Spanish naturalist and court physician Francisco Hernández de Toledo (1514–1587) [33] (p. 927). Accompanied by his son, Hernández researched the plants of Mexico for their medicinal properties. It can therefore be assumed that he observed the indigo production process during this expedition (1570–1577). However, neither Jörlin (1759) nor Ray (1686) provide information on the country where the indigo production took place. The production is described as follows: The leaves are cut and put into a pot of boiling water, and after some time, the pot is taken off the fire to cool. To bring oxygen into the solution, shovels are moved vigorously. Then the solution is transferred into a vessel with holes higher up. After the sediment has sunk, the water leaves the vessel through the holes. The indigo is dried in the sun and small indigo balls are formed. Jörlin cites Hermann who reported that a blue dye, superior to indigo, is made from *Tephrosia tinctoria* by the Sinhalese but hitherto unknown to the Europeans [34] (p. 302).

Tournesol or lacmus originating from the dyer's croton (*Chrozophora tinctoria*), native to the Mediterranean region, Central Asia and Northwest India, provides a blue color for dyeing paper and pharmaceuticals [35]. As the color is easily changed by alkali and acid, it is also applied to test the water quality. Linnaeus used the dyer's croton when he tested the quality of a well at Roma Monastery on Gotland in 1741 [19] (p. 293), [20] (p. 175).

The juice of the flowers of the field larkspur (*Delphinium consolida*), boiled with alum, gives a blue color for scribes; the corollas of the harebell (*Campanula rotundifolia*) and the cornflower (*Centaurea cyanus*) are other sources for writing. The bark of the common ash (*Fraxinus excelsior*) turning blue in water is specified in a recipe by Linder: A cloth is pretreated with the ground cedar (*Lycopodium complanatum*) and then dyed blue with the inner bark of the common ash (*Fraxinus excelsior*) [23] (p. 63). Interestingly, American settlers took the inner bark of the blue ash (*Fraxinus quadrangulate* Michx.) to dye yarns blue [36]. The yellow flowers of the woundwort (*Anthyllis vulneraria*) and the bird's foot clover (*Lotus corniculatus*) become blue after drying. From these color observations, Jörlin concludes that these flowers are suitable for dyeing blue.

**Table 9.** Dyeing materials for blue.

| Grou p | Jörlin -No. | Name in Jörlin<br>*Linnaean Name* | Material | English Name<br>*Swedish Name* | *Current Scientific Name*<br>Family |
|---|---|---|---|---|---|
| A | 51<br>51* | DELPHINIUM Consolida<br>*Delphinium consolida*<br>Linnaeus | Flowers | Field larkspur<br>*Riddar-sporre* | *Delphinium consolida* L.<br>Ranunculaceae |
| A<br>B | 58<br>58* | ISATIS tinctoria<br>*Isatis tinctoria* Linnaeus | Herb | Woad<br>*Weide, (vejde)* | *Isatis tinctoria* L.<br>Brassicaceae |
| B | 59<br>59* | INDIGOFERA tinctoria<br>*Indigofera tinctoria* Linnaeus | Leaves | Indigo<br>*Indigo* | *Indigofera tinctoria* L.<br>Fabaceae |
| B | 85<br>85* | CROTON tinctorium<br>*Croton tinctorius* Linnaeus | Fruits | Dyer s croton, folium<br>*Lacmus, tournesol* | *Chrozophora tinctoria* (L.) A.Juss.<br>Euphorbiaceae |
| C | 18<br>17* | CAMPANULA rotundifolia<br>*Campanula rotundifolia*<br>Linnaeus | Flowers | Harebell, Scottish bluebell<br>*Kläcka, (liten blåklocka)* | *Campanula rotundifolia* L.<br>Campanulaceae |
| C | 59,61<br>- | ANTHYLLIS Vulneraria<br>*Anthyllis vulneraria* Linnaeus | Flowers | Woundwort, Kidneyvetch<br>*Rafklor, (getväppling)* | *Anthyllis vulneraria* L.<br>Fabaceae |
| C | 59<br>- | Lotus corniculate<br>*Lotus corniculatus* Linnaeus | Flowers | Bird s-foot trefoil<br>*(Käringtand)* | *Lotus corniculatus* L.<br>Fabaceae |
| C | 72<br>- | CENTAUREA Cyanus<br>*Centaurea cyanus* Linnaeus | Flowers | Cornflower<br>*Blåklint* | *Centaurea cyanus* L.<br>Asteraceae |
| C | 90<br>90* | FRAXINUS excelsior<br>*Fraxinus excelsior* Linnaeus | Bark | Common ash<br>*Ask* | *Fraxinus excelsior* L.<br>Oleaceae |
| D | 60<br>60* | GALEGA tinctoria<br>*Galega tinctoria* (Linnaeus) Linnaeus | Leaves | Orange Tephrosia<br>- | *Tephrosia tinctoria* (L.) Pers.<br>Fabaceae |

Swedish names between brackets are from other sources than Jörlin; * Number occurring in the index sorted by color in Jörlin [3] (p. 342); A = native plant, used in Sweden; B = imported trade product; C = native plant with potential use for dyeing; D = non-native plant, used abroad.

### 9. Greens—*Viridis*

For dyeing green, ten materials are listed (Table 10). The herb of two plants, the panicles of two grasses, flowers of five species contain anthocyanins and the unripe alder buckthorn fruits provide flavonoid dyes.

In the general part of the dissertation, Jörlin mentions that green is created by dyeing yellow and blue, but nevertheless ten species are given. The unripe fruits of the alder buckthorn (*Frangula alnus*) dye wool in a green color. Swedish countrymen apply the panicles of the common reed (*Phragmites australis*). In Linnaeus' opinion, two plant species were known for dyeing green in Scania, the panicles of the rye brome (*Bromus secalinus*) by the farmers near the city of Ystad, and the flower heads of the red clover (*Trifolium pratense*) for woolen garments [26] (p. 277).

The flowers of the German bearded iris (*Iris germanica*), boiled with alum, and the juice of squeezed flowers of the pasqueflower (*Anemone pulsatilla*) yield green tinctures for writing. Woolen clothes can be dyed a saturated green with the roots, stems and leaves of the ragwort (*Jacobaea vulgaris*). They are picked before flowering, not dried and boiled with the textiles. However, the color is weakened by the solar rays ("*color autem, a radiis solaribus debilitatur*"). Pure and beautiful greens are obtained with the herb of the cow parsley (*Anthriscus sylvestris*), the freshly squeezed flowers of the common bugloss (*Anchusa officinalis*) and the flowers of the harebell (*Campanula rotundifolia*), the last two are boiled with alum.

**Table 10.** Dyeing materials for green.

| Group | Jörlin-No. | Name in Jörlin<br>*Linnaean Name* | Material | English Name<br>*Swedish Name* | *Current Scientific Name*<br>Family |
|---|---|---|---|---|---|
| A | 6<br>6* | BROMUS secalinus<br>*Bromus secalinus* Linnaeus | Panicles | Rye brome<br>*Losta, Swingel* | *Bromus secalinus* L.<br>Poaceae |
| A | 7<br>7* | ARUNDO phragmites<br>*Arundo phragmites* Linnaeus | Panicles | Common reed<br>*Was, (bladvass, rörvass)* | *Phragmites australis* (Cav.) Trin. ex Steud.<br>Poaceae |
| A | 19<br>- | RHAMNUS frangula<br>*Rhamnus frangula* Linnaeus | Berries | Alder buckthorn<br>*Tröske, (brakved)* | *Frangula alnus* Mill.<br>Rhamnaceae |
| A | 63<br>63* | TRIFOLIUM pratense<br>*Trifolium pratense* Linnaeus | Flowers | Red clover<br>*Rodwapling, (rödklöver)* | *Trifolium pratense* L.<br>Fabaceae |
| C | 14<br>14* | ANCHUSA officinalis<br>*Anchusa officinalis* Linnaeus | Flowers | Common bugloss<br>*Oxtunga* | *Anchusa officinalis* L.<br>Boraginaceae |
| C | 18<br>- | CAMPANULA rotundifolia<br>*Campanula rotundifolia* Linnaeus | Flowers | Harebell<br>*Kläcka, (liten blåklocka)* | *Campanula rotundifolia* L.<br>Campanulaceae |
| C | 22<br>22* | CHAEROPHYLLUM sylvestre<br>*Chaerophyllum sylvestre* Linnaeus | Herb | Cow parsley<br>*Hundkaxa* | *Anthriscus sylvestris* (L.) Hoffm.<br>Apiaceae |
| C | 52<br>52* | ANEMONE Pulsatilla<br>*Anemone pulsatilla* Linnaeus | Flowers | Pasqueflower<br>*Backsippa* | *Anemone pulsatilla* L.<br>Ranunculaceae |
| C | 69<br>39* | SENESIO Jacobaea<br>*Senecio jacobaea* Linnaeus | Herb with roots | Ragwort<br>*Stånds* | *Jacobaea vulgaris* Gaertn.<br>Asteraceae |
| D | 4<br>4* | IRIS germanica<br>*Iris germanica* Linnaeus | Flowers | German bearded iris<br>*Swerdslilja* | *Iris germanica* L.<br>Iridaceae |

Swedish names between brackets are from other sources than Jörlin; * Number occurring in the index sorted by color in Jörlin [3] (p. 342); A = native plant, used in Sweden; C = native plant with potential use for dyeing; D = non-native plant, used abroad.

## 10. Purples—*Purpurei*

For dyeing purple, ten materials are listed (Table 11). Molluscan purple provides 6,6′-dibromoindigotin and other indigoid dyes, three species of soluble redwoods yield the homoisoflavonoid dye brazilein, berries of two species and flowers of one species contain anthocyanins. The following three sorts of soluble redwoods were imported: sappanwood from Asia, brasilletto from Central America and pernambuco wood from Brazil.

Jörlin writes that the purple of antiquity, once sung by the Greeks and Romans, was produced from a certain snail species. He mistakenly thinks that the snail gives off a liquid when pierced between its tentacles but knows very well that the pale liquid of the snail turns purple when exposed to the sun. He mentions *Turbo clathrus* as a possible species but is not sure that it is the right source. Whatever its origins, Jörlin states that molluscan purple is little missed because American cochineal (*Dactylopius coccus*) provides a beautiful color. Nowadays, it is known that real purple does not originate from this marine snail species but from the banded dye-murex the banded dye-murex, *Hexaplex trunculus* (Linnaeus, 1758), the spiny dye-murex, *Bolinus brandaris* Linnaeus, 1758, and the red-mouthed rockshell, *Stramonita haemastoma* (Linnaeus, 1767).

Three imported species of soluble redwoods are given: pernambuco wood (*Paubrasilia echinata*), sappanwood (*Biancaea sappan*) and brasiletto (*Caesalpinia vesicaria*). They give beautiful reds and purples, mainly for dyeing cloths. For pernambuco he points to Ray's *Historia Plantarum* [33] (p. 1737) and to the *Historia Plantarum Universalis* written by the Swiss botanists and physicians Johan Bauhin (1541–1613) and Johann Heinrich Cherler (1569–1609/10) [37] (p. 494).

The privet berries (*Ligustrum vulgare*) provide a purple dye and in his *Historia Plantarum*, Ray notes that these berries serve to illuminate playing cards and other pictures ("*qui chartulas lusorias aliasve picturas variis coloribus illumninant*") [33] (p. 1603). Black crowberries (*Empetrum nigrum*), boiled with alum, dye clothes a dark purple hue. Referring to Linnaeus, Jörlin states that farmers in the province Scania and on the island Öland gain purple colors from the flowering branches of the oregano (*Origanum vulgare*) [19] (pp. 58, 97, 101), [26] (p. 277). During the Öland journey, he observed that oregano created reds near Isgärde, brown colors in the neighborhood of Hulterstad and brownish reds in the area of Sandby [20] (pp. 53, 73, 76).

**Table 11.** Dyeing materials for purple.

| Group | Jörlin-No. | Name in Jörlin<br>*Linnaean Name* | Material | **English Name**<br>*Swedish Name* | *Current Scientific Name*<br>Family |
|---|---|---|---|---|---|
| A | 2<br>- | LIGUSTRUM vulgare<br>*Ligustrum vulgare*<br>Linnaeus | Berries | Wild privet<br>*Liguster* | *Ligustrum vulgare* L.<br>Oleaceae |
| A | 55<br>55* | Origanum vulgare<br>*Origanum vulgare* Linnaeus | Flowers | Oregano<br>*Dosta, (Kungsmynta)* | *Origanum vulgare* L.<br>Lamiaceae |
| B | 36<br>36* | CAESALPINIA vesicaria<br>*Caesalpinia vesicaria*<br>Linnaeus | Wood | Brasilletto<br>*Brasilletta* | *Tara vesicaria* (L.) Molinari, Sánchez Och.<br>& Mayta; *Caesalpinia vesicaria* L.<br>Fabaceae, subfamily Caesalpinioideae |
| B | 37<br>37* | CAESALPINIA Sappan<br>*Caesalpinia sappan*<br>Linnaeus | Wood | Sappanwood<br>*Sappan* | *Biancaea sappan* (L.) Tod.,<br>syn. *Caesalpinia sappan* L.<br>Fabaceae, subfamily Caesalpinioideae |
| B | 38<br>38* | LIGNUM rubrum<br>- | Wood | Pernambuco wood,<br>brazilwood<br>*Fernbock* | *Paubrasilia echinata* (Lam.) Gagnon,<br>H.C.Lima & G.P.Lewis, *Caesalpinia echinata* Lam.<br>Fabaceae, subfamily Caesalpinioideae |

| Grou, | Jörlin-No. | Name in Jörlin / Linnaean Name | Material | English Name / Swedish Name | Current Scientific Name / Family |
|---|---|---|---|---|---|
| C | 87 / 87* | EMPETRUM nigrum [1] / *Empetrum nigrum* Linnaeus | Berries | Black crowberry / *Kraklinge, (Kråkbär, kråkris)* | *Empetrum nigrum* L. / Empetraceae |
| D | 107 / - | TURBO Clathrus / *Turbo clathrus* Linnaeus, 1758 | Snails | - / *Purpur* | *Epitonium clathrus* (Linnaeus, 1758) / Epitoniidae |

Swedish names between brackets are from other sources than Jörlin; * Number occurring in the index sorted by color in Jörlin [3] (p. 342); [1] Jörlin mentions that the berries provide a dark purple color, but in the index, it is only placed for dyeing violet (*violace*i); A = native plant, used in Sweden; B = imported trade product; C = native plant with potential use for dyeing; D = non-native animal with historical use abroad.

## 11. Violets—*Violacei*

For dyeing violet, four materials are listed (Table 12). The flowers of two species and berries of one species contain anthocyanins and logwood the homoisoflavonoid dye haematein. The following two materials were imported: the Malabar spinach from tropical Asia and logwood from Central America.

Jörlin reports that the juice from the petals of the common violet (*Viola odorata*) is applied for dyeing pharmaceuticals and to detect acid and alkali in water. Imported logwood (*Haematoxylum campechianum*), a tree native to the Central America (*arbor Indiae occidentalis*), gives a violet hue on cloths but mainly serves as a base for black. The juice of the berries of the Malabar spinach (*Basella alba*), which grows in tropical Asia, dyes an elegant violet color, but due to its poor colorfastness it is only suitable for cosmetics, tinctures and clothlets (*bezetta*). The flowers of the black vanilla orchid (*Nigritella nigra*) add a violet shade to the spirit of corn.

**Table 12.** Dyeing materials for violet.

| Grou, | Jörlin-No. | Name in Jörlin / Linnaean Name | Material | English Name / Swedish Name | Current Scientific Name / Family |
|---|---|---|---|---|---|
| A | 75 / 74* | VIOLA odorata / *Viola odorata* Linnaeus | Flowers | Common violet / *Fioler, (luktviol)* | *Viola odorata* L. / Violaceae |
| B | 34 / 34* | HAEMATOXYLON campechianum / *Haematoxylum campechianum* Linnaeus | Wood | Logwood, Campeche wood / *Blauholtz, Campecheträd* | *Haematoxylum campechianum* L. / Fabaceae, subfamily Caesalpinioideae |
| B | 26 / 26* | BASELLA rubra [1] / *Basella rubra* Linnaeus | Berries | Malabar spinach / *(Malabarspenat)* | *Basella alba* L. / Basellaceae |
| C | 75 / 75* | SATYRIUM nigrum / *Satyrium nigrum* Linnaeus | Flowers | Black vanilla orchid / *Brunkulla* | *Nigritella nigra* (L.) Rchb. / Orchidaceae |

Swedish names between brackets are from other sources than Jörlin; * Number occurring in the index sorted by color in Jörlin [3] (p. 342); [1] Jörlin mentions that this plant provides a fugitive violet, but in the index, it is only placed for dyeing red (*rubri*); A = native plant, used in Sweden; B = imported trade product; C = native plant with potential use for dyeing.

## 12. Blacks—*Nigri*

For dyeing black, ten materials are listed (Table 13). Barks, leaves, fruit cups and galls contain mainly tannins, gypsywort and berries of foreign plants possess unknown dyes. The cups of the valonia oak were imported from the Mediterranean region.

Jörlin mentions that the leaves of the bearberry (*Arctostaphylos uva-ursi*) are often used for dyeing, and Linnaeus described this plant growing in Lapland, Småland

(Brånsmåla, Villeköl) and Gotland (Martebo) [31] (pp. 122–125), [19] (pp. 21, 32, 182), [20] (pp. 33, 39, 119). In a cloth factory in Norrköping, Linnaeus was told that a black was dyed with the bearberry twigs and vitriol instead of foreign sumac (*Rhus*) [19] (p. 10), [20] (p. 28). The branches and leaves of the bearberry, known as mjölon or 'Swedish sumac', dyed alum-mordanted wool yellow, and an iron-based mordant was used to obtain dull yellow, green, brown and grey tints. Farmers from all parts of Sweden sold this product to the cities, where it was applied by tanners and dyers for green, grey and 'castor black'. Mjölon was exported to Finland and known by dyers in Norway and Denmark [38].

The alder bark (*Alnus incana*) provides a red color, but in particular fishermen dye their nets black using this bark with *martialia* (iron-containing agents). Bark, galls and fruit cups of oaks (*Quercus* sp.) are other sources. Oak galls, rich in tannins, form on the underside of oak leaves after the female gall wasps (*Cynips quercusfolii*) have laid their eggs in the leaves. The common oak (*Quercus robur*) is known as the tree of iron-black (*arbor nigrum Martiale*). The imported cups of the valonia oak (*Quercus ithaburensis*), native to the Mediterranean region, are more common among tanners than among dyers. The juice of baneberries (*Actaea spicata*) boiled with alum provides a black ink. The 'Cingali' dye their faces brown to black with the juice of gipsywort (*Lycopus europaeus*), which cannot be washed away. Two other plant species also dye the skin. When eaten, the berries of various *Melastoma* species, native to Asia, dye the mouth and lips black, which lasts for a fortnight and the same is true for the unripe berries of *Genipa americana*. Hands washed in this juice get a color that cannot be removed for fourteen days. American Indians apply these berries in two ways, to dye their clothes and to paint their faces with the juice to deter their enemies.

**Table 13.** Dyeing materials for black.

| Grou | Jörlin -No. | Name in Jörlin *Linnaean Name* | Material | English Name *Swedish Name* | *Current Scientific Name* Family |
|---|---|---|---|---|---|
| A | 39 39* | ARBUTUS Uva ursi *Arbutus uva-ursi* Linnaeus | Leaves | Bearberry *Mjölonris, (mjölon)* | *Arctostaphylos uva-ursi* (L.) Spreng. Ericaceae |
| A | 78 - | BETULA Alnus *Betula alnus* Linnaeus | Bark | Black alder and grey alder *Al* | *Alnus glutinosa* (L.) Gaertn. and *Alnus incana* (L.) Moench; Betulaceae |
| A | 82 82* | QUERCUS robur *Quercus robur* Linnaeus | Fruits | Common oak *Ek* | *Quercus robur* L. Fagaceae |
| A | 82 82* | QUERCUS robur *Quercus robur* Linnaeus | Bark | Common oak *Ek* | *Quercus robur* L. Fagaceae |
| A | 100 100* | CYNIPS Qvercus folii *Cynips quercusfolii* Linnaeus, 1758 | Galls | Oak galls *Galläple* | *Cynips quercusfolii* Linnaeus, 1758 Cynipidae |
| B | 83 83* | QUERCUS Aegilops *Quercus aegilops* Linnaeus | Fruits | Valonia oak *(Valonia-ek)* | *Quercus ithaburensis* Decne. ssp. macrolepis (Kotschy) Hedge & Yalt. Fagaceae |
| C | 3 3* | LYCOPUS europaeus *Lycopus europaeus* Linnaeus | Herb | Gipsywort *(Strandklo)* | *Lycopus europaeus* L. Lamiaceae |
| C | 48 | ACTAEA spicata | Berries | Baneberry | *Actaea spicata* L. |

| | | | | |
|---|---|---|---|---|
| | 48* | *Actaea spicata* Linnaeus | *Trollbär*, (*trolldruva*) | Ranunculaceae |
| D | 17<br>17* | GENIPA americana<br>*Genipa americana* Linnaeus | Berries | -<br>- | *Genipa americana* L.<br>Rubiaceae |
| D | 40<br>40* | MELASTOMA variarum<br>specierum<br>- | Berries | -<br>- | *Melastoma malabathricum* L. and<br>other species of Melastomaceae |

Swedish names between brackets are from other sources than Jörlin; * Number occurring in the index sorted by color in Jörlin [3] (p. 342); A = native plant, used in Sweden; B = imported trade product; C = native plant with potential use for dyeing; D = non-native plant, used abroad.

### 13. Lichens

Seven lichen species are listed for dyeing purple and yellow (Table 14). Lichens that contain depsides and depsidones provide the purple dye orcein. The yellow colors are due to anthraquinone or xanthone dyes.

Lichens played an important role in the Swedish dye history especially during the Age of Utility (1719–1771), when the use of domestic manufactured products was encouraged. Their application has been subject of investigation by Linder, Linnaeus and two of his students, Pehr Kalm and Johan Peter Westring.

An article written by Kalm [27], and the dissertation of his student Carl Fridric Leopold *Korta frågor angående nyttan af inländska water*, 1753) (*Short questions concerning the use of our domestic plants*) [39] mention native dye lichens, stating that "no realm has more of these [lichens] than Sweden" [40] (p. 299). The Swedish physician and naturalist Johan Peter Westring (1753–1833) investigated the dyeing properties of numerous lichens and published his studies in Swedish in 1805 [41]. It was also his idea to replace expensive imports of dyeing materials with native species to benefit the Swedish economy.

Jörlin writes that *byttelet* (*Ochrolechia tartarea*) gives a purple color and is often used by the inhabitants of Västergötland [32] (pp. 146–147). This lichen species was gathered by coastal dwellers and fishermen along the west coast of Sweden and Norway. Especially after rainfall, it was scraped off the cliffs with the help of a sharp bent knife or a special tool. Linnaeus was told that the inhabitants of the town Borås travelled annually to the Hisingen archipelago north of Gothenburg to collect these lichens. They prepared *Böttelet*, also named *Boråsfärg* (Borås dye), which peddlers traded all over the country. Merchants in Gothenburg bought large quantities of this product and exported it to Scotland for dyeing tweed [40] (p. 301), [21] (p. 158). Kalm described the harvesting and dyeing of *Ochrolechia tartarea* (*Lichen leprosus candidus*) in Bohuslän, a province on the west coast of Sweden: The lichens were scraped off the rocks immediately after rainfall. Then they were put into water and stirred up to remove small stones and impurities, which sank to the bottom of the vessel. Next, the lichens were dried in the sun, ground and washed again. The clean powder was dried, put into a jug with urine and left for one month. When the dye was needed, a spoonful or more was taken and put into a vessel with water. After quickly boiling the cloth was dyed, taken out and finally hung up to dry [27] (pp. 245–246), [40] (p. 298).

Another source is crottle (*Parmelia saxatilis*), which is popular for dyeing wool and clothes in rural areas, as Linnaeus observed in Småland, on Öland and Gotland [19] (pp. 30, 101, 209), [20] (pp. 38, 75, 132). In Sandby on Öland, he saw the dyeing process of wool with 'stone moss' (*Parmelia saxatilis*): The lichens and unmordanted yarns were placed layer by layer in a pot and then boiled in water with lye, a strong alkaline solution made by soaking wood ash in water [19] (p. 101), [20] (p. 75). Without additives, the lichens gave the yarns beautiful brown ('moss brown') to brownish-red colors, which was practiced all over the country [40] (p. 103), [21] (p. 157). It seems that only native lichens were applied to dye textiles. Interestingly, orchil (*Roccella tinctoria*), growing mainly on rocks in the

Mediterranean area, is not imported to Sweden for textile dyeing, but Jörlin mentions its use to make the wine brighter in color.

Four lichen species are the sources for yellow dyes. Swedish women take *Xanthoria parietina* and Swedish countrymen dye woolen clothes with *Vulpicida juniperinus,* while *Xanthoria candelaria* and *Trentepohlia jolithus* are of less importance.

**Table 14.** Lichens for purple and yellow.

| Group | Jörlin-No. | Name in Jörlin<br>*Linnaean Name* | Color | English Name<br>*Swedish Name* | *Current Scientific Name*<br>Family |
|---|---|---|---|---|---|
| A | 92<br>- | LICHEN tartareus<br>*Lichen tartareus* Linnaeus | Purple | Viking dye-moss<br>*Byttelet* | *Ochrolechia tartarea* (L.) A.Massal.<br>Ochrolechiaceae |
| A | 93<br>93* | LICHEN saxatilis<br>*Lichen saxatilis* Linnaeus | Purple | Crottle<br>*Stenmoßa* | *Parmelia saxatilis* (L.) Ach.<br>Parmeliaceae |
| B | 92<br>92* | LICHEN Roccella<br>*Lichen roccella* Linnaeus | Purple | Orchil lichen<br>*Orseille* | *Roccella tinctoria* DC.<br>Roccellaceae |
| A | 94<br>94* | LICHEN Juniperinus [(1)]<br>*Lichen juniperinus* Linnaeus | Yellow | -<br>*Enmoßa (enlav)* | *Vulpicida juniperinus* (L.) J.-E. Mattsson & M. J. Lai<br>Parmeliaceae |
| A | 95<br>- | LICHEN parietinus<br>*Lichen parietinus* Linnaeus | Yellow | Yellow wall lichen<br>*Wäggmåssa* | *Xanthoria parietina* (L.) Th. Fr.<br>Teloschistaceae |
| A | 96<br>- | LICHEN candelarius<br>*Lichen candelarius* Linnaeus | Yellow | -<br>*Liusmåssa, (ljus-laf)* | *Xanthoria candelaria* (L.) Th.Fr.<br>Teloschistaceae |
| A | 97<br>97* | BYSSUS Jolithus<br>*Byssus jolithus* Linnaeus | Yellow | -<br>*Violmåssa* | *Trentepohlia jolithus* (L.) Wallr.<br>Trentepohliaceae |

Swedish names between brackets are from other sources than Jörlin; * Number occurring in the index sorted by color in Jörlin [3] (p. 342); [(1)] Jörlin mentions this lichen only for dyeing yellow, but in the index, it is also noted for dyeing purple (*purpurei*); A = native plant, used in Sweden; B = imported trade product.

## 14. Conclusions

The hitherto hardly known dissertation *Plantae tinctoriae* by Engelbert Jörlin can be regarded as an historical document on dyeing materials from the mid-18th century. The thesis is successfully studied by applying the 'Noscemus General Model' from Transkribus to transcribe the Latin text, creating a Microsoft Access database and consulting several websites for the information on dye plants and dye insects.

40% of the used dye materials are of native origin (A), 20% are imported (B), 28% of the materials are native to Sweden having a potential to be used in dyeing (C), and 12% concern non-native species that are used abroad (D). During the Age of Utility (1719–1771), the use of domestic manufactured products was encouraged to replace expensive imported trade goods. Native lichens played an important role in that period. One aim of the dissertation was to find new indigenous dyeing materials as substitutes.

The materials were mainly applied for dyeing textile materials and textiles, like wool, silk, woolen cloths, clothes and garments but also for other applications, such as coloring pharmaceuticals, cosmetics (make-up), tinctures and spirit of corn. Red and violet clothlets (*bezetta*), inks and the artist's pigments *Schutgelb* (Stil de grain jaune) and *Sutgrön* (Stil de grain vert) were used for art purposes. The fruits of dyer's croton (*Chrozophora tinctoria*) and the petals of the common violet (*Viola odorata*) served for testing the water quality because their dyes are sensitive to acid and alkali. Jörlin is aware that some dyes have a

poor lightfastness, such as the green from ragwort (*Jacobaea vulgaris*), the red from dyer's alkanet (*Alkanna matthioli*), the violet from berries of the Malabar spinach (*Basella alba*) and the pink of inkberries (*Phytolacca americana*).

He performed dyeing experiments to test the dyeing properties of certain plants but he does not go into detail. Without knowing it, Jörlin describes the anthocyanins in an indirect way. He mentions the characteristic color change of the blue dyes from flowers and fruits into red with acid and green with alkali (*colores caerulei, ab acido rubri, ab alcali virides evadunt*). However, it was not until 1835 that the German pharmacist Ludwig Clamor Marquart (1804–1881) gave the name *Anthokyan* to a chemical compound from blue flowers in his treatise *Die Farben der Blüthen*. The knowledge derived from Jörlin's research is now accessible to the scientific community and maybe there are more secrets hidden in his dissertation.

**Author Contributions:** Conceptualization, R.H.-d.K. and M.d.K.; Methodology, R.H.-d.K. and M.d.K.; Writing—Original draft, R.H.-d.K. and M.d.K.; Writing—Review & editing, R.H.-d.K. and M.d.K. All authors have read and agreed to the published version of the manuscript.

**Funding:** This research received no external funding.

**Data Availability Statement:** All data are provided in the manuscript.

**Acknowledgments:** The authors would like to express their gratitude to Robert Kralofsky in Vienna, Austria, who carried out the transcription of Jörlin's dissertation with the 'Noscemus General Model' from Transkribus and designed a Microsoft Access database for the dyeing materials mentioned by Jörlin. The authors sincerely acknowledge Jo Kirby Atkinson in London, Great Britain, for her contribution to historical colorants and Ines Bogensperger in Salzburg, Austria, for the translation and interpretation of ancient Greek and Latin texts. Special thanks go to the following persions and institutions for providing images and information: Charlotte Tancin, librarian at the Hunt Institute for Botanical Documentation, Carnegie Mellon University, Pittsburgh, PA, USA; the Swedish Literature Bank, https://litteraturbanken.se/om/inenglish (accessed on 23 November 2022); the Digitalisierungsplattform der Zentralbibliothek Zürich e-rara.ch; Stefan Zathammer, creator of the 'Noscemus General Model' from Transkribus, Noscemus Read&Search, Innsbruck, Austria; and the Niedersächsische Staats- und Universitätsbibliothek Göttingen, Germany.

**Conflicts of Interest:** The authors declare no conflict of interest.

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
