# Peer review of "Plantae tinctoriae: The 1759 Dissertation on Dye Plants by Engelbert Jörlin"

_heritage, doi:10.3390/heritage6020081_

Round 1
Reviewer 1 Report
This is a really interesting paper; you are certainly to be congratulated for bringing Jörlin's thesis to the attention of the dyeing community, and to a wider readership as well. I think there are a few points that need attention, the most important of which is that you need a conclusion to draw the threads together.
Lines 77, 78 onwards: Investigative methods. It might help the non-botanists among your readers if you include a sentence, probably here, to explain that since Linnaeus compiled his taxonomy, plant classification has moved on so that plants (and scale insects) have been re-examined and in some cases, reassigned to other families and/or renamed.
Line 93: This is Figure 3, not 2.
In general: You need to make it absolutely clear throughout your discussion that you are describing, discussing or explaining what Jörlin is saying, as opposed to what the position is today, so where you are explaining the modern position you may need have a word or two to indicate this. Otherwise the reader is going to get confused.
Lines 123, 134: Jörlin's discussion of colour. This might work better if you link the first sentence with the next paragraph by taking the sentence about Newton out of lines 123-4 and put it with the creation of colours; you will need to adjust the wording slightly. The paragraph beginning at line 123 will then be on the use of dyes.
Lines 180, 181 and following: Do you have references for Dioscorides and Pliny the Elder? Both are available online in various editions.
Lines 235, 241: Sap green, vert de vessie in French, is not strictly a lake pigment as it is not precipitated onto a substrate: the sticky juice of ripe buckthorn berries is expressed and stored in pieces of pig's bladder until needed; usually a little alum is added. The sugars naturally present in the juice acts as a binding medium and the pigment is used as a watercolour. (Flavonoids + anthocyanidin colorants, these from the purple skins, give the colour.)
Line 286: I know you mention lady's bedstraw later under reds, obtained from the roots, but I wonder if a note ('see below for the red obtained from the roots') might be a good idea in case the reader is puzzled about this plant being mentioned under yellows (I'm amazed it was used in this way; the inflorescence of this plant is tiny!)
Lines 343, 344: Safflower. Interesting that the presence of the yellow colorant (would this not interfere?) is not mentioned, also that the red is actually alkali soluble but precipitated from alkaline solution by acid. Maybe this needs some explanation? Did Jörlin actually try to do this?
Line 448, Table 8: Dragon's blood. There are several sources for this, but I assume that Linnaeus and Jörlin only mentioned this one?
Lines 580 and 591, Table 13 no. 83: Please clarify the way the oak galls are produced: the wasp injecting the egg causes the development of the gall (the tannins are produced by the tree).
Line 629: Unmordanted is the word you need.
Line 643: Conclusion please!!
Author Response
Dear Reviewer,
Thank you very much for your fruitful comments. You will find our responses in the attached file.
Kind regards
Regina Hofmann-de Keijzer and Matthijs de Keijzer

Reviewer 2 Report
In the abstract the authors explains the aims of the article. This is not followed up in the introductory parts, and needs to be explained further.
The research is very interesting, and can be used by many scholars and researchers interested in dyes from a scientifically, artistically, historically or makers/ crafters point of view. This should be mentioned, and also something about further research.
The authors could improve the readability, by clearly point out the different time layers in the article. One example is on line 249–250, about experiments on dyeing. My understanding is that this is experiments made recently, but it could be understood as an experiment made historically. If my understanding is correct, references are needed.
A conclusion should be added to the article, explaining the results by reconnecting to the aims. Here the subject of further research based on the results, should be raised.
Author Response

(The authors gave the same response as above.)

Reviewer 3 Report
This is an excellent examination and review of a dissertation prepared in the mid 18th century by a student of the globally-renowned Swedish botanist Carl Linnaeus on the use of natural dyes in Sweden at the time, both indigenous and imported. The thesis was digitised, translated from Latin, and the content critically analysed in an updated context. This paper will make an excellent contribution to knowledge on historical use of natural dyes.
Minor comment: change 1750ies to 1750s.
Author Response

(The authors gave the same response as above.)

Reviewer 4 Report
Manuscript ID: heritage-2113361
Type of manuscript: Article
Title: Plantae tinctoriae: the 1759 dissertation on dye plants by Engelbert Jörlin
1. Recommendation:
Accept after minor revisions
2. Comments to the authors:
2.1 Overview and general recommendation
The paper "Plantae tinctoriae: the 1759 dissertation on dye plants by Engelbert Jörlin" is an interesting revision of a historical record of dyeing plants used in Sweden. In my opinion, the authors present a very interesting historical document that can contribute to the knowledge of dye sources used in the past, therefore, I think it can considerably contribute to several fields (e.g., Art history, Heritage Science, Conservation) interested on studying dyes and textile production.
As a general concern, sometimes is difficult to understand when the text is a paraphrasis or modernization of Jörlin’s text and when is information added by the authors. The text would benefit from making clearer the difference between the information from the original text and the authors' contribution. In this respect, citations to support some chemical information regarding the dyes mentioned should be added. Moreover, there are no conclusions, authors should add some concluding remarks to the text discussing the possible implications of this paper in the investigation of dyeing sources and textile production.
For these reasons, I recommend accepting the paper after minor revisions. I explain my suggestions in detail below.
2.2 Comments
Page 4. Figure 2. There is no y-axis in the bar chart, please add it and specify the units. Also, the number is wrong. It should be Figure 3.
Page 4. Figure 4. If possible, please substitute the image with another with higher quality and bigger to make it easier to read the text.
Section 3. Please elaborate a little more about the provenance of the imported materials. Maybe a graph or a table summarizing the places (e.g., countries or continents) from where the materials were imported to Sweden can be useful to correlate this information with the geo-economic/political situation of the time.
Page 5. Line 107. Probably the term “bar chart” is more adequate instead of “diagram”
Page 5. Line 128. You mention: “[…] many books of famous authors, […]” please consider moving Table 2 to this place since it might be correlated.
Page 5. Line 140. Does the color change happen to the plant or to the dye obtained from it?
Page 6. Line 187. is La Mancha instead of Lamancha.
Page 7. Line 198-99. Did Jörlin indicate the dye chemical class in his text (e.g., Flavonoids) or is it an addition by the authors? If it is the last, please add a citation to support the information regarding the dye classes. This should be done for the other classes. In the same lines is mentioned that saffron contains crocetin, it actually contains some other dyes, such as crocin, please add more precise information.
Page 9. Table 3. Please add the meaning of the group letters in the caption of the table, this can help the reader avoids going back to Figure 3 to understand the meaning.
Page 9. Line 258. Is the term “milk” used by Jörlin or is it by the authors? I suggest putting it “milk” if it comes from the original text or using only latex – a more technical term – if it is a modern contribution.
Page 15. Line 354. You should place Table 17 after its first citation in the text. Please move it to this place.
Page 22. Line 523. You should place Table 11 after its first citation in the text. Please move it to this place.
Page 23. Line 553. You should place Table 12 after its first citation in the text. Please move it to this place.
Page 24. Line 567. You should place Table 13 after its first citation in the text. Please move it to this place.
Page 25. Lines 595-597. Please add citations to support this information.
Please add conclusion remarks.
Author Response

(The authors gave the same response as above.)
